# The Dystrophin-Dystroglycan complex ensures cytokinesis efficiency in *Drosophila* epithelia

Margarida Gonçalves [1,2,3], Catarina Lopes[1,2], Hervé Alégot[4], Mariana Osswald[1,2], Floris Bosveld [5], Carolina Ramos[1,2], Graziella Richard[4], Yohanns Bellaiche [5], Vincent Mirouse [4] & Eurico Morais-de-Sá [1,2✉]

## Abstract

**Cytokinesis physically separates daughter cells at the end of cell division. This step is particularly challenging for epithelial cells, which are connected to their neighbors and to the extracellular matrix by transmembrane protein complexes. To systematically evaluate the impact of the cell adhesion machinery on epithelial cytokinesis efficiency, we performed an RNAi-based modifier screen in the *Drosophila* follicular epithelium. Strikingly, this unveiled adhesion molecules and transmembrane receptors that facilitate cytokinesis completion. Among these is Dystroglycan, which connects the extracellular matrix to the cytoskeleton via Dystrophin. Live imaging revealed that Dystrophin and Dystroglycan become enriched in the ingressing membrane, below the cytokinetic ring, during and after ring constriction. Using multiple alleles, including Dystrophin isoform-specific mutants, we show that Dystrophin/Dystroglycan localization is linked with unanticipated roles in regulating cytokinetic ring contraction and in preventing membrane regression during the abscission period. Altogether, we provide evidence that, rather than opposing cytokinesis completion, the machinery involved in cell–cell and cell–matrix interactions has also evolved functions to ensure cytokinesis efficiency in epithelial tissues.**

**Keywords** Cell Adhesion; Cytokinesis; Dystrophin; Epithelial Tissues; Extracellular Matrix
**Subject Categories** Cell Adhesion, Polarity & Cytoskeleton; Cell Cycle

## Introduction

Cytokinesis, the physical separation of daughter cells, relies on a dramatic remodeling of the cytoskeleton. It begins with the assembly of an actomyosin ring at the cell equator, which drives membrane ingression, and is completed by the abscission of a small intercellular bridge, or midbody, formed at the end of ring constriction. Cytokinesis has long fascinated researchers, leading to the identification of conserved structural components and regulators (Pollard and O'Shaughnessy, 2019). These include formins, which drive F-actin polymerization, non-muscle Myosin II, which drives actomyosin contraction, and anillin and septins, which tether the ring to the plasma membrane. In addition, cytokinetic ring contraction must be adapted to different animal cell types (Cabernard et al, 2010; Davies et al, 2018; Jordan et al, 2016; Ozugergin and Piekny, 2022; Paim and FitzHarris, 2022). A particularly interesting case is that of epithelia, which must maintain tissue cohesion and the epithelial barrier throughout cell division. Here, dividing cells face challenges posed by cell–cell and cell–extracellular matrix (ECM) interactions (Daniel et al, 2018; Herszterg et al, 2014; Higashi et al, 2016; McKinley et al, 2018; Osswald and Morais-de-Sa, 2019; Wang et al, 2018). Understanding these challenges is essential, as cytokinesis failure leads to genome duplication and centrosome amplification, which are key contributors to oncogenesis (Lens and Medema, 2019; Levine et al, 2017; Vittoria et al, 2023).

Previous studies of epithelial cytokinesis focused on aspects relevant for animal tissues topology and morphogenesis: E-cadherin-based adherens junctions (AJ) remodeling, the coordination with de novo junction formation and the role of AJ in the direction of ring closure (di Pietro et al, 2023; Firmino et al, 2016; Founounou et al, 2013; Guillot and Lecuit, 2013; Herszterg et al, 2013; Morais-de-Sa and Sunkel, 2013; Pinheiro et al, 2017). However, few studies have yet examined the impact of the epithelial context on the efficiency of cytokinesis completion. Work in the *Drosophila* pupal notum epithelium has shown that in the absence of septins, AJ promote cytokinesis failure (Founounou et al, 2013). In line with this, increased tension at AJ, due to tight junction defects or excessive neighbor cell contractility, led to cytokinesis failure in *Xenopus* epithelia (Hatte et al, 2018; preprint: Landino et al, 2023). Cell–matrix interactions may also interfere with epithelial cytokinesis. Accordingly, in the zebrafish epicardium, focal adhesions that connect the cytokinetic ring to the ECM promote cytokinesis failure when reinforced (Uroz et al, 2019). Thus, while cell adhesion molecules are not inherently essential for

[1]IBMC - Instituto de Biologia Molecular e Celular, Universidade do Porto, 4200-135 Porto, Portugal. [2]Instituto de Investigação e Inovação em Saúde (i3S), Universidade do Porto, 4200-135 Porto, Portugal. [3]Programa Doutoral em Biologia Molecular e Celular (MCBiology), Instituto de Ciências Biomédicas Abel Salazar, Universidade do Porto, Porto, Portugal. [4]Université Clermont Auvergne - iGReD (Institute of Genetics, Reproduction and Development), UMR CNRS 6293 - INSERM U1103, Faculté de Médecine, Clermont-Ferrand, France. [5]Institut Curie, Université PSL, Sorbonne Université, CNRS UMR3215, INSERM U934, Genetics and Developmental Biology, 75005 Paris, France. ✉E-mail: eurico.sa@ibmc.up.pt

cytokinesis, they can potentially interfere with cytokinesis efficiency, threatening organismal homeostasis.

The follicular epithelium of the *Drosophila* ovary combines genetic tractability with the power to image epithelial cytokinesis in an adult organ. We designed an RNAi-based genetic modifier screen to test the impact of the major regulators of cell–cell and cell–matrix interactions on cytokinesis efficiency. This uncovered unexpected cytokinetic functions for the Dystrophin–Dystroglycan complex, a transmembrane ECM–cytoskeleton linker with important implications in neuromuscular dystrophies and cancer (Jones et al, 2021; Mirouse, 2023; Nowak and Davies, 2004).

## Results and discussion

### *Drosophila* modifier screen to test the impact of cell–cell and cell–matrix interactions on cytokinesis efficiency

We performed an RNAi modifier screen in the follicular epithelium to identify modulators of epithelial cytokinesis efficiency. To produce a modifiable phenotype with the mild frequency of multinucleated cells, we targeted Anillin, a component of the cytokinetic ring that tethers it, as well as the midbody ring to the plasma membrane (Kechad et al, 2012; Zhang and Maddox, 2010). To ensure temporally controlled and tissue-specific expression of Anillin RNAi, we induced UAS-driven Anillin RNAi with the follicular epithelium-specific *tj-GAL4* driver (Olivieri et al, 2010) and blocked the GAL4 transcription factor with its temperature-sensitive repressor Gal80[ts] (McGuire et al, 2003) until 3 days prior to tissue dissection (Fig. 1A). We marked the plasma membrane (Myr:GFP) and nuclei (DAPI) to enable automated quantification of the multinucleation ratio (nuclei/cell) in stage 10 egg chambers (Fig. EV1) as a proxy for cytokinesis failure during earlier proliferative stages. In these conditions, Anillin RNAi led to mild multinucleation, which is suitable to monitor rescue or enhancement of defects.

To validate the strategy to identify modifiers of cytokinesis efficiency, we tested the impact of reinforcing apical cell adhesion. Overexpression of E-Cadherin (ECad) led to a significant increase in multinucleation (Fig. 1B,C), consistent with earlier findings that AJ challenge cytokinesis (Founounou et al, 2013). We then used RNAi lines to test the impact of the cadherin superfamily members expressed in the follicular epithelium (based on the modENCODE transcriptional profile (Brown et al, 2014)). Controls co-expressing UAS-Anillin RNAi and UAS-mCherry were processed in parallel to each independent experiment, and multinucleation for each RNAi was compared to its respective control (Δ multinucleation ratio). Intriguingly, ECad depletion had no effect on multinucleation (Fig. 1D), whereas depleting N-cadherin (NCad) reduced it. This suggests that NCad depletion, but not ECad, effectively reduces apical cell adhesion in follicle cells, which is likely explained by NCad compensating for ECad loss during the proliferative stages ((Tanentzapf et al, 2000) and Fig. EV3A). Depletion of the apical adhesion molecules Cad74A and Cad99C also reduced the multinucleation ratio (Fig. 1D). In contrast, depletion of the non-canonical cadherin Fat2, *kugelei* (*kug/fat2*), enriched at the basal side (Viktorinova et al, 2009), or of Calsyntenin (Cals), with unknown localization, increased the multinucleation ratio (Fig. 1D). Altogether, this suggested that in the follicular epithelium cadherin-

based adhesion only opposes cytokinesis completion when exerted at the apical level.

A second large family of cell adhesion molecules belongs to the immunoglobulin superfamily (Finegan and Bergstralh, 2020). This includes Echinoid (Ed), the functional orthologue of mammalian Nectin, which forms a second adhesion complex at AJ, as well as Neuroglian (Nrg), Fasciclin 2 (Fas2), Fasciclin 3 (Fas3), Lachesin (Lac) and Contactin (Cont), which localize to the lateral membrane in the follicular epithelium prior to septate junction maturation. We did not observe any effect of depleting Fas2, Fas3, and Lac. However, Cont RNAi suppressed multinucleation, whereas Nrg and Ed depletion enhanced it (Fig. 1E). Nrg interacts heterotypically both with Ed and the single-pass transmembrane protein Neurexin IV (NrxIV) (Banerjee et al, 2006; Islam et al, 2003). NrxIV RNAi also increased multinucleation, suggesting Nrg-NrxIV and Nrg-Ed complexes in the lateral membrane promote epithelial cytokinesis efficiency.

Integrins are heterodimeric receptors formed by α and β subunits that work as physical linkers between the cytoskeleton and the ECM (Kanchanawong and Calderwood, 2023). Knockdown of the only β subunit in follicle cells, Myospheroid (*mys*/βPS), or of the α subunits Scab (*scb*/αPS3) and Multiple edematous wings (*mew*/αPS1) did not affect multinucleation (Fig. 1F). To corroborate this conclusion, we validated integrin protein depletion and its functional impact by reproducing its reported effect on egg chamber elongation (Fig. EV3B, (Qin et al, 2017)). Interestingly, knocking down the non-integrin ECM receptor Dystroglycan (*Dg*) increased multinucleation, suggesting it plays a positive role in cytokinesis (Fig. 1F).

The mechanical properties of the matrix and its ability to signal via focal adhesions modulate cytokinesis efficiency in cell culture (Rabie et al, 2021; Sambandamoorthy et al, 2015). We postulated that ECM composition could impact cytokinesis efficiency and screened the main ECM components in the basement membranes of follicle cells: Collagen IV (*viking*/Col4α2), Laminins, Perlecan (*trol*) and Nidogen (Diaz-Torres et al, 2021). Depletion of Collagen IV, the major regulator of basement membrane stiffness in the follicular epithelium (Crest et al, 2017; Topfer et al, 2022), caused previously reported egg chamber elongation defects (Fig. EV3B), but did not impact multinucleation (Figs. 1G and EV3B). Laminin B2 (LanB2) depletion reduced multinucleation, but RNAi for Laminin A (LanA) or Laminin B1 (LanB1) did not modify the multinucleation ratio, despite of an efficient reduction of protein levels (Fig. EV2I,J). In contrast, Perlecan depletion increased multinucleation dramatically, suggesting it promotes cytokinesis robustness (Fig. 1G).

Altogether, the genetic modifier screen indicates that whereas AJ can challenge epithelial cytokinesis efficiency, there are a number of transmembrane proteins involved in cell–cell and cell–matrix interactions that rather promote efficient cytokinesis (screen data overview in Dataset EV1; validation of protein depletion to confirm the main screen results in Fig. EV2). It is worth noting that depletion of Ed, Nrg, NrxIV, Dg, Kug, or Perlecan did not produce multinucleated cells on their own (Fig. EV3C), which suggests these proteins are dispensable for cytokinesis but contribute to cytokinesis efficiency.

### The Dystrophin-associated protein complex promotes cytokinesis efficiency

The importance of regulating cell–substrate interactions during cytokinesis has been well-studied in cell culture (Taneja et al, 2019),

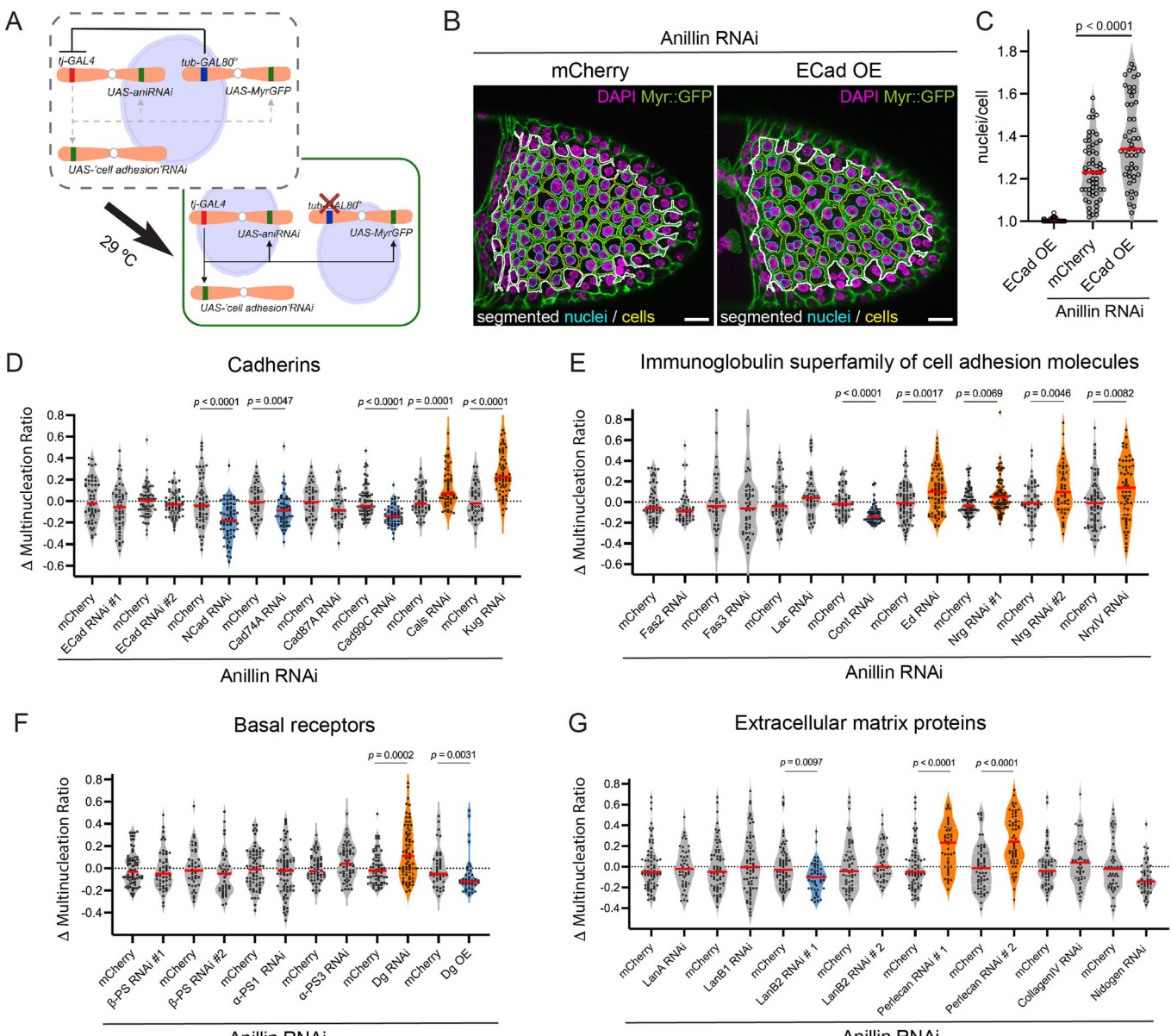

**Figure 1. Impact of cell adhesion molecules on epithelial cytokinesis efficiency.**

(A) Genetic strategy of the RNAi modifier screen in the follicular epithelium. (B) Automated segmentation of cells (yellow) and nuclei (cyan) in the central area (white line) of egg chambers expressing Anillin RNAi simultaneously with a control UAS-mCherry transgene or overexpressing E-cadherin. Myr:GFP marks cell membranes; DAPI labels nuclei. Scale bars: 20 μm. (C) Nuclei/cell in stage 10 egg chambers expressing UAS-Ecad on its own ($n = 53$) or co-expressing Anillin RNAi with UAS-ECad ($n = 52$), or with UAS-mCherry ($n = 59$). (D–G) Modification of the Anillin RNAi multinucleated cell phenotype by co-depletion of proteins from the Cadherin (D) and Immunoglobulin superfamily (E), basal receptors (F), and extracellular matrix components (G). Enhancers (orange) and suppressors (blue) are highlighted. UAS-mCherry was used as a control (Δ multinucleation ratio = (nuclei/cell)$_{RNAi}$ − mean (nuclei/cell)$_{mCherry}$). (C–G) Each dot represents an egg chamber (median in red). *P* value calculated by non-parametric unpaired Mann–Whitney test. Source data are available online for this figure.

but remains unexplored in epithelia. We therefore addressed the role for the basal ECM receptor Dg in cytokinesis efficiency. Dg overexpression reduced cytokinesis defects caused by Anillin RNAi (Fig. 1F). In addition, Dg RNAi enhanced the multinucleation frequency produced by depletion of Tumbleweed (human RacGAP1, Fig. EV3D), a centralspindlin complex component that regulates multiple aspects of cytokinesis (White and Glotzer, 2012). These findings further support the positive role of Dg in cytokinesis.

Dg is a transmembrane heterodimeric component of the Dystrophin-associated protein complex (DAPC). It links the ECM to the intracellular cytoskeleton by interacting with Dystrophin (Dys), a cytoplasmic actin-binding protein (Fig. 2A). To investigate if Dys also modulates epithelial cytokinesis, we assessed the genetic interaction between Anillin RNAi and *Dys* mutant alleles. We used a genomic deletion (*Df (3R)Exel6184*, named *Dys^Df^*) that spans the entire *Dys* locus, along with *Dys^E17^* (nonsense mutation resulting in a truncated

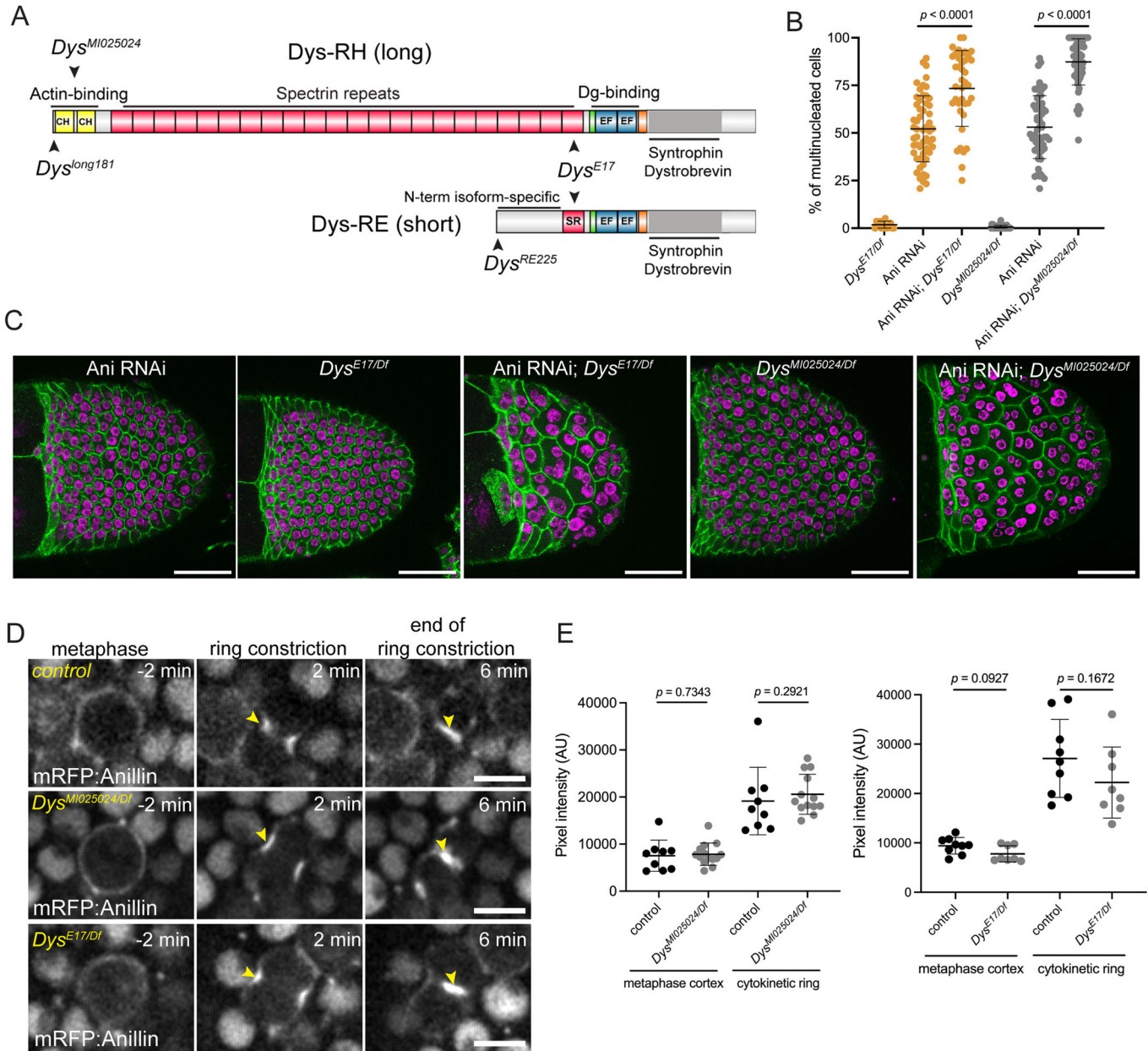

**Figure 2. Dystrophin promotes cytokinesis efficiency.**

(**A**) Schematic representation of Dys isoforms. The position of the insertion mutant Dys^{MI025024} and the truncating mutations Dys^{E17} (deletes the Dg-binding domain in both isoforms), Dys^{long181} (disrupts long isoform) and Dys^{RE225} (disrupts short isoform) is shown. (**B, C**) Frequency of multinucleated cells in egg chambers from Anillin RNAi, Dys^{E17/Df} and Dys^{MI025024/Df} mutants with unperturbed or sensitized (Anillin RNAi) cytokinesis. (**B**) Graph shows mean ± SD. Each dot represents an egg chamber. P value calculated by non-parametric unpaired Mann–Whitney test. (**C**) Surface projections show cell membranes (ECad:GFP) and nuclei (DAPI). Scale bars: 50 μm. (**D**) Surface projections of dividing control, Dys^{MI025024/Df} and Dys^{E17/Df} follicle cells expressing mRFP:Anillin. Arrowheads indicate Anillin enrichment during or after ring constriction. Scale bars: 5 μm. (**E**) Graph shows mean (±SD) pixel intensity of mRFP:Anillin measured in the metaphase cortex (2 min before initiation of ring constriction and in the cytokinetic ring (2 min after constriction onset). Each dot represents a dividing cell. n_{control} = 9; n_{DysMI025024/Df} = 14; n_{control} = 9; n_{DysE17/Df} = 8. P value calculated by non-parametric unpaired Mann–Whitney test. Source data are available online for this figure.

protein without the Dg-binding domain) or with Dys^{MI025024} (*minos* insertion in the exon that encodes the actin-binding domain; Fig. 2A). Neither Dys^{E17/Df} nor Dys^{MI025024/Df} egg chambers exhibited a significant number of multinucleated cells, but *Dys* mutant alleles in the background of Anillin depletion led to a dramatic increase in multinucleated cells when compared to Anillin RNAi alone (Fig. 2B,C).

A potential role of Dys on Anillin recruitment to the contractile ring could explain this phenotypic enhancement. However, mRFP:Anillin redistribution during cytokinesis was unaffected by loss of *Dys* function (Fig. 2D,E). We conclude that the DAPC promotes cytokinesis efficiency but does not directly contribute for Anillin cytokinetic recruitment.

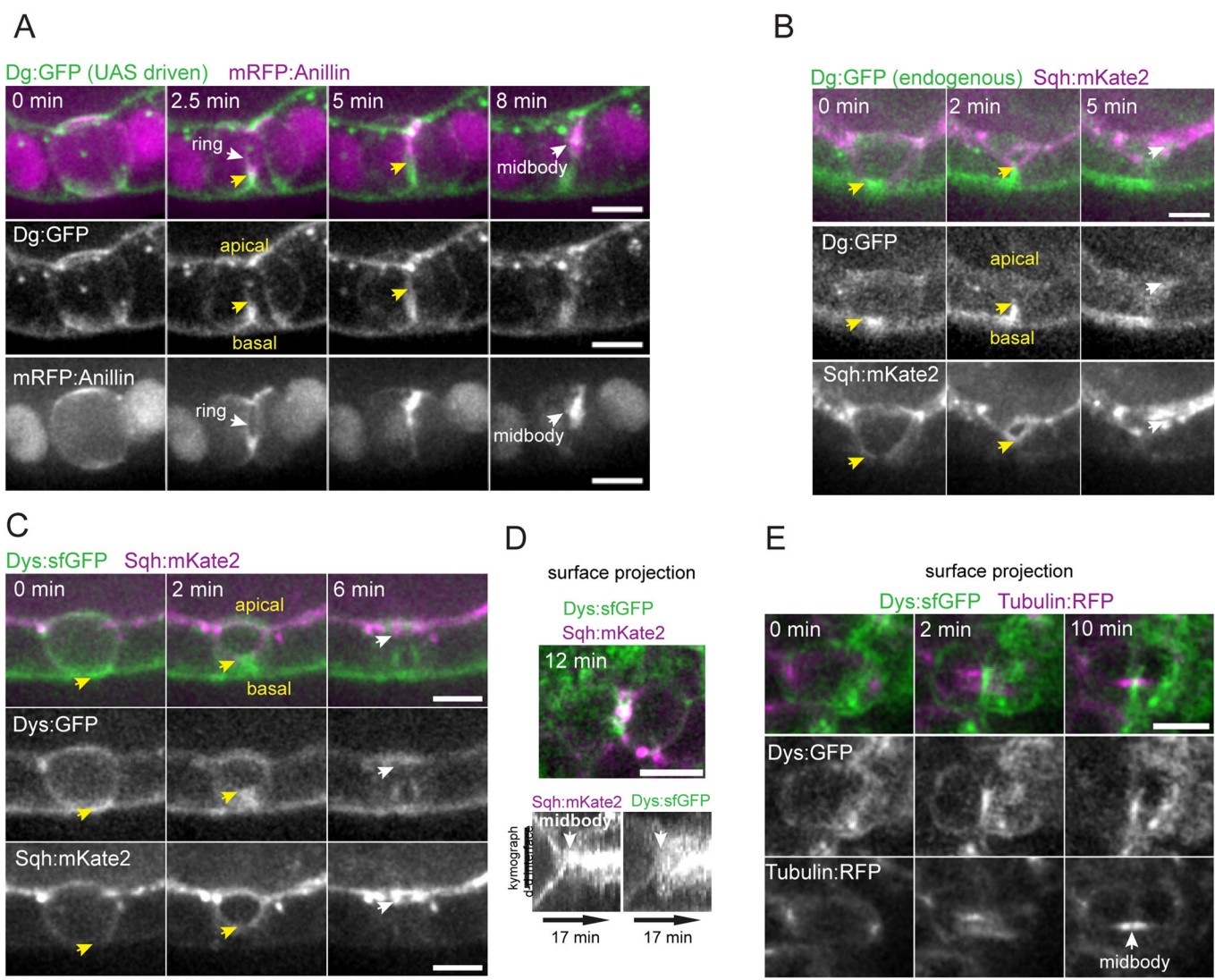

**Figure 3. DAPC redistribution during cytokinesis.**

(A–E) Time-lapse images of follicle cells expressing UAS-driven Dg:GFP (**A**) or endogenously tagged Dg:GFP (**B**) and Dys:sfGFP (**C–E**). Anillin:mRFP (**A**), Sqh:mKate2 (**B–D**) or Tub:RFP (**E**) label the contractile ring and the midbody. Apical-basal views (**A–C**) show that (**A**) UAS-driven Dg:GFP becomes enriched in the ingressing furrow (yellow arrows). Endogenous Dg:GFP (**B**) and Dys:sfGFP (**C**) accumulate at the basal part of the ingressing membrane from the start of ring constriction (yellow arrows) and become enriched close to the midbody (white arrows). (**D**) A line kymograph (bottom) along the daughter–daughter (d–d) interface from a surface projection (top) shows that upon midbody formation (arrow), Dys:sfGFP maintains an accumulation at the new d–d interface. (**E**) Surface projection showing the accumulation of Dys:sfGFP near the midbody-associated microtubules (arrow). Scale bars: 5 μm. Source data are available online for this figure.

## Dynamic DAPC redistribution during cytokinesis

To investigate how the DAPC functions during epithelial cytokinesis, we examined its distribution in dividing follicle cells expressing UAS-driven Dg:GFP and mRFP:Anillin, which labels first the cytokinetic ring and then the midbody as it is assembled at the end of constriction (Fig. 3A). Dg:GFP strongly accumulated at the ingressing membrane below the cytokinetic ring while it constricts towards the apical domain until midbody formation. Overexpressed Dg:GFP was not restricted to the basal side in interphase cells, unlike observations by antibody staining (Schneider et al, 2006). To rule out overexpression effects, we examined the

mitotic redistribution of endogenously tagged Dg:GFP and Dys:sfGFP (Fig. 3B,C; Movies EV1–EV3). These became locally enriched at the basal part of the membrane at the onset of constriction and accumulated at the ingressing membrane below the ring during cytokinesis. After ring constriction, Dys:sfGFP and Dg:GFP accumulated at the new interface between daughter cells, close to the midbody (marked by Sqh:mKate2 upon constriction (Fig. 3D) or by midbody-associated microtubules labeled with Tubulin:RFP (Fig. 3E)). The cytokinetic redistribution of these proteins is not common to all cell types, as the main Dys isoforms do not show any cortical accumulation during cytokinesis in *Drosophila* S2 cells (Fig. EV4B). Nevertheless, the DAPC may have

cytokinetic functions beyond epithelia, as Dys and Dg showed cleavage furrow enrichment in rat fibroblasts and non-polarized cell culture models (Higginson et al, 2008; Villarreal-Silva et al, 2011). More importantly, in the *Drosophila* pupal notum epithelium, Dg and Dys also accumulated strongly at the ingressing membranes just below the contractile ring and persisted post-constriction along these membranes below the midbody ring (Fig. EV4C,D, respectively). Thus, the Dg–Dys complex redistributes dynamically during cytokinesis in different epithelial tissues, in line with a role during epithelial cytokinesis.

Dys encodes several isoforms in both humans and flies. All of them bear a conserved C-term region with the Dg-binding domain (Fig. 2A), but the shorter isoforms lack actin-binding domains (N-term actinin-like domain and a second actin-binding domain within a rod domain). Translating ribosome affinity purification experiments show that only one long (Dys-RH) and one short isoform (Dys-RE) are expressed in the follicular epithelium (preprint: Vachias et al, 2023). To investigate whether Dys redistribution and function during cytokinesis rely on direct binding to the actin cytoskeleton, we generated by CRISPR/Cas9 isoform-specific indel mutant alleles in an untagged *Dys* genomic locus (*Dys^long181^*, i.e., mutating all long isoforms including RH, and *Dys^RE225^* mutating RE) and in a sfGFP-tagged one. Naming of *Dys^long181^*:sfGFP and *Dys^RE225^*:sfGFP was simplified by indicating the tagged isoforms: *Dys^short^*:sfGFP and *Dys^long^*:sfGFP, respectively. Western blot confirmed the long isoform's absence in *Dys^short^*:sfGFP and the short isoform's absence in *Dys^long^*:sfGFP (Fig. 4A). Both isoforms showed strong basal enrichment in the proliferative follicular epithelium, even if *Dys^short^*:sfGFP exhibited a more uniform distribution in the basal cortex that contrasts with *Dys^long^*:sfGFP planar polarization (Fig. EV5A,B). More importantly, both isoforms accumulated below the ingressing furrow during constriction, and at the new interface post-constriction, suggesting both contribute to epithelial cytokinesis (Fig. 4B; Movie EV4). Accordingly, *Dys* disruption with either isoform-specific untagged mutant alleles (*Dys^long181^* and *Dys^RE225^*) dramatically increased multinucleation upon Anillin depletion (Fig. 4C,D). Thus, both Dys isoforms, regardless of the actin-binding domains, localize at the ingression furrow and contribute to cytokinesis efficiency.

To identify protein domains involved in Dys enrichment at the ingressing furrow, we generated UAS-driven *Dys^short^*:GFP truncations that retained the ability to bind Dg, but lacked evolutionarily conserved domains that are common to all isoforms: the N-term spectrin repeat (*Dys^shortΔSR24^*) or a motif in the C-terminal part that interacts with cytoplasmic adaptors, syntrophins and Dystrobrevin (*Dys^shortΔSD^*). *Dys^shortΔSR24^* partially mislocalized to the apical side and nuclei (Fig. EV5C), but it was still enriched at the ingressing membranes during ring constriction and upon midbody formation (Fig. EV5E). In contrast, *Dys^shortΔSD^* had reduced basal enrichment and did not accumulate in the ingressing membranes during cytokinesis (Fig. EV5C,F). This effect requires the removal of the syntrophins/Dystrobrevin binding sites, as a shorter deletion of the most C-terminal region (aa956-1051, *Dys^shortΔCT^*) did not affect Dys cytokinetic redistribution (Fig. EV5G). Syntrophins and Dystrobrevin function in epithelial tissues has yet to be understood. Our results show that the cortical and cytokinetic localization of Dys relies on motifs that interact with these scaffolding proteins, a region containing an alpha-helix and adjacent coiled-coil domains (Newey et al, 2000; Sadoulet-Puccio et al, 1997).

## Multiple roles of the DAPC complex on epithelial cytokinesis efficiency

Anillin is required for the maturation of the midbody ring, that stabilizes the intercellular bridges during abscission. Accordingly, Anillin RNAi in *Drosophila* S2 cells leads to reopening of the furrow a significant period of time after ring constriction (Kechad et al, 2012). To further understand the genetic interaction between Anillin RNAi and *Dys* in cytokinesis, we monitored follicle cell cytokinesis by live imaging (Fig. 5A). We could barely detect cytokinesis failure upon Anillin depletion on its own, but additional disruption of *Dys* function increased the frequency of cytokinesis failure (Fig. 5C). Interestingly, almost all cells reached a semi-stable state after ring closure, but the new interface formed between the daughter cells regressed as late as 30 min post-ring constriction (Fig. 5B,C; Movie EV5), suggesting the DAPC contributes to the efficiency of the last stage of cytokinesis.

Since Dys and Dg spatially redistribute in the furrowing membrane during cytokinesis, we also postulated that the DAPC regulates early cytokinetic furrowing. We therefore monitored cytokinetic ring constriction in *Dg* (*Dg^043/086^*) and *Dys* (*Dys^E17/Df^*) egg chambers. *Dg* and *Dys* disruption significantly delayed ring constriction (Fig. 5D,E; Movie EV6) from the beginning of furrowing, as the ring diameter at the onset of the phase of constant ring constriction (1.5 min) is already significantly different. To specifically examine the impact of long and short *Dys* isoforms, we also imaged egg chambers mutant for each isoform (*Dys^RE225/Df^* or *Dys^long181/Df^*). Both mutants impaired cytokinetic closure (Fig. 5F). In addition, *Dys* isoforms and *Dg* also contributed to normal ring constriction rate during the phase of constant ring constriction (Fig. 5G). Hence, we conclude the DAPC ensures normal ring constriction in the follicular epithelium.

In conclusion, this study provides new insights into the mechanisms that ensure epithelial cytokinesis efficiency, and uncovers new cytokinetic functions for the DAPC in *Drosophila* tissues. Previous in vivo studies have shown that cell–cell and cell–matrix adhesion could potentiate cytokinesis failure and delay ring constriction (Founounou et al, 2013; Hatte et al, 2018; Higashi et al, 2016; Uroz et al, 2019). Our RNAi modifier screen now identifies a number of cell–cell and cell–matrix interaction mediators that promote, rather than oppose, cytokinesis completion. These include Perlecan, a Dg ligand (Talts et al, 1999), Fat2, a genetic interactor of Dys (Cerqueira Campos et al, 2020), and cell–cell adhesion molecules from the immunoglobulin superfamily. Building on this screen, we show that the DAPC, through long and short *Dys* isoforms, functions during cytokinetic furrowing and contributes to efficient daughter cell separation after ring constriction.

How the DAPC contributes to each step of cytokinesis remains an important open question. We show that Dg and Dys accumulate below the contractile ring during furrowing, which may contribute to remodel cell–matrix interactions to facilitate basal plasma membrane invagination. Indeed, the DAPC contributes to the dynamic organization of the basement membrane and underlying cytoskeleton in different morphogenetic processes (Buisson et al, 2014; Cerqueira Campos et al, 2020; Villedieu et al, 2023). DAPC enrichment in late cytokinesis near the midbody ring suggests it could also regulate the abscission machinery. Interestingly, Dys binds microtubules and pauses their polymerization (Belanto et al, 2014; Prins et al, 2009), so it could be involved in the disassembly of

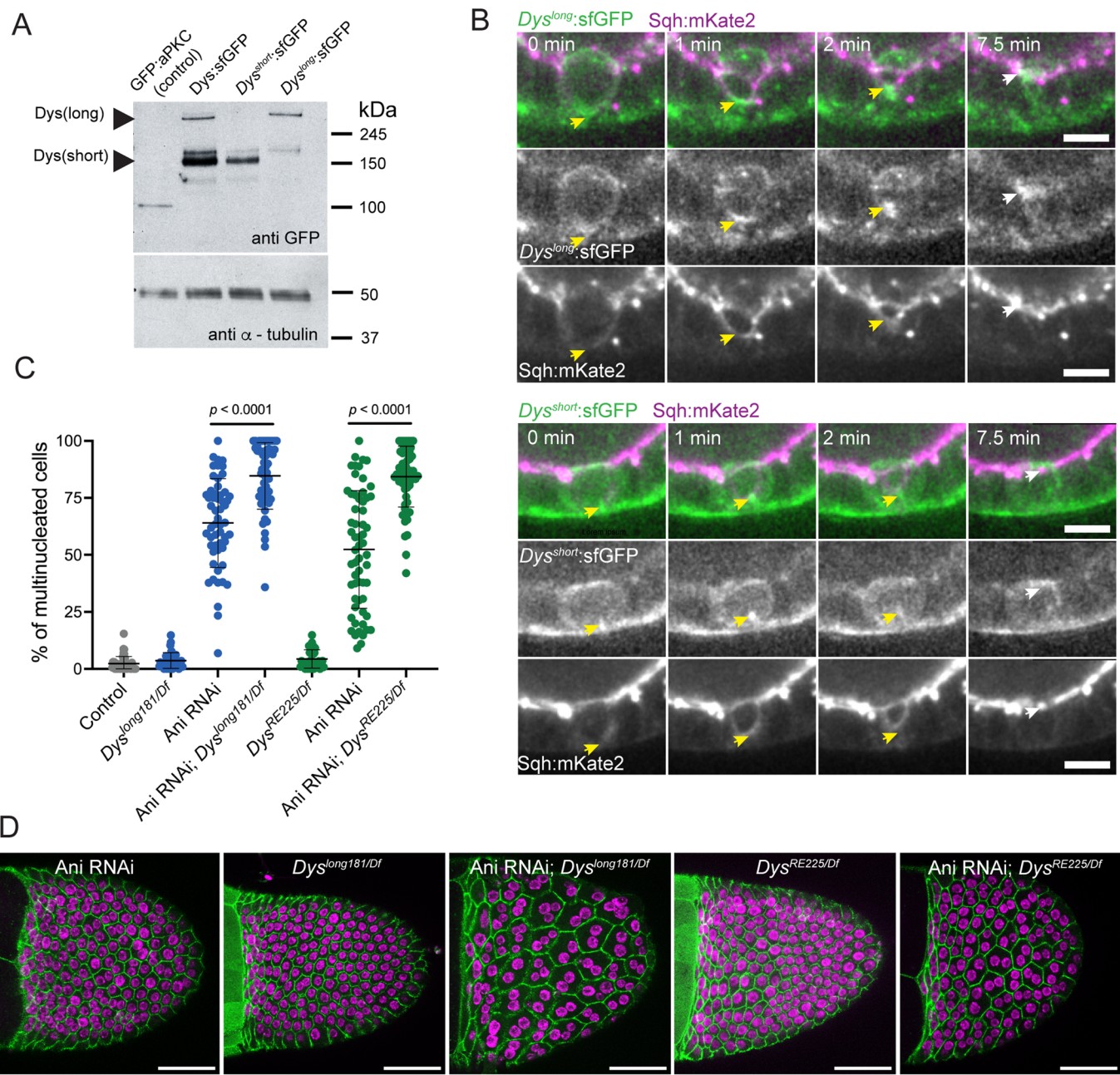

**Figure 4. Short and long Dys isoforms contribute for cytokinesis efficiency.**

(A) Western blots from ovaries of GFP:aPKC (GFP detection control), Dys:sfGFP; $Dys^{short}$:sfGFP and $Dys^{long}$:sfGFP flies probed with anti-GFP or anti-α-tubulin. (B) Endogenously tagged $Dys^{long}$:sfGFP (top) and $Dys^{short}$:sfGFP (bottom) accumulate at the basal ingressing membrane from the start of ring (labeled with Sqh:mKate2) constriction (yellow arrows) and become enriched close to the apically positioned midbody (white arrows). (C, D) Frequency of multinucleated cells in control, $Dys^{long181/Df}$ and $Dys^{RE225/Df}$ egg chambers with unperturbed or sensitized (Anillin RNAi) cytokinesis. Graph shows mean ± SD. Each dot represents an egg chamber. P value calculated by non-parametric unpaired Mann–Whitney test. (D) Surface projections show cell membranes (ECad:GFP) and nuclei (DAPI). Scale bars: 50 μm. Source data are available online for this figure.

midbody-associated microtubules. Alternatively, the DAPC could facilitate cytokinesis completion by linking the cell cytoskeleton to the ECM to enhance cell–matrix adhesion. This would be consistent with cell culture studies suggesting that maintaining adhesion to the substrate facilitates cytokinesis in cells with a compromised actomyosin ring (Dix et al, 2018). Although Dys

cytokinetic function requires isoforms lacking known actin-binding domains, adaptor proteins that interact with conserved Dys C-terminal motifs could link the DAPC to the cytoskeleton. Future work can focus on these hypotheses to elucidate how the DAPC coordinates the intracellular cytoskeleton and the extracellular ECM to promote cytokinesis fidelity in different tissues.

# Methods

## Reagents and tools table

| Reagent/resource | Reference or source | Identifier or catalog number |
|---|---|---|
| **Experimental models** | | |
| *Drosophila* (*D. melanogaster*) | List of transgenic lines and alleles is shown in Table EV1/EV2 | N/A |
| S2 cells (*D. melanogaster*) | *Drosophila* Genomics Resource Center | FlyBase: FBtc0000179 |
| **Recombinant DNA** | | |
| *DysEcDNA* | *Drosophila* GOLD cDNA collection | RE11449 |
| *pHWG* | *Drosophila* Genomics Resource Center | Gateway Collection |
| *pH-DysE:GFP* | This study | N/A |
| *pCFD6* | Port and Bullock, 2016 | Cat# AddGene-73915 |
| *pUASz* | DeLuca and Spradling, 2018 | Cat# DGRC-1431 |
| *pUAS-Dys$^{short}$* | This study | N/A |
| *pUAS-Dys$^{short\Delta SD}$* | This study | N/A |
| *pUAS-Dys$^{short\Delta SR24}$* | This study | N/A |
| *pUAS-Dys$^{short\Delta CT}$* | This study | N/A |
| *pUAS-GFP:DysRH* | This study | N/A |
| *pActinGal4* | Potter et al, 2010 | Cat# AddGene 24344 |
| **Antibodies** | | |
| Rat anti-N-cadherin | DSHB | Cat# DN-Ex #8 |
| Mouse anti-Fas2 | DSHB | Cat# 1D4 |
| Mouse anti-Fas3 | DSHB | Cat# 7G10 |
| Mouse anti-βPS-integrin | DSHB | Cat# CF.6G11-s |
| Rrabbit anti-Perlecan | González-Reyes lab (Diaz-Torres et al, 2021) | N/A |
| Mouse anti-Armadillo | DSHB | Cat# N2.7A1 |
| Goat anti-rat Alexa 568 | Invitrogen | Cat# A11077 |
| Goat anti-mouse Alexa 488 | Invitrogen | Cat# A11029 |
| Goat anti-rabbit Alexa 488 | Invitrogen | Cat# A11008 |
| Goat anti-mouse Alexa 568 | Invitrogen | Cat# A11031 |
| Rabbit anti-GFP | i3S core facility | N/A |
| Goat anti-rabbit HRP | Jackson ImmunoResearch | Cat# 111-035-003 |
| **Oligonucleotides and other sequence-based reagents** | | |
| gRNAs | This study | Methods (Section "Molecular biology and transgenesis") |
| **Chemicals, enzymes, and other reagents** | | |
| Phalloidin-TRITC | Merck | Cat# P1951 |
| Vectashield Mounting Medium with DAPI | Vector Laboratories | Cat# H-1200-10 |
| CellMask Orange Plasma Membrane Stain | Thermo Fisher | Cat# C10045 |

| Reagent/resource | Reference or source | Identifier or catalog number |
|---|---|---|
| Poly-D-lysine | Sigma-Aldrich | Cat# P7405 |
| Schneider's insect medium | Sigma-Aldrich | Cat# S0146 |
| Insulin solution from bovine pancreas | Sigma-Aldrich | Cat# I0516 |
| Fetal Bovine Serum (FBS), heat-inactivated | Thermo Fisher | Cat# 10500-064 |
| Paraformaldehyde 20% | Electron Microscopy Sciences | Cat# 15713 |
| Tween 20 | Sigma-Aldrich | Cat# P9416 |
| Effectene Transfection Reagen | QIAGEN | Cat# 301425 |
| NEB Builder HiFi DNA Assembly Cloning Kit | New England Biolabs | Cat# E5520S |
| LR Clonase II | Thermo Fisher | Cat# 11791020 |
| **Software** | | |
| FIJI | Schindelin et al, 2012 | N/A |
| GraphPad Prism 9 | GraphPad Software (La Jolla, CA, USA) | N/A |
| **Other** | | |
| Laser Scanning Confocal Microscope TCS SP5 II | Leica Microsystems | N/A |
| Spinning disk confocal microscope Andor Revolution XD | Andor Technology | N/A |

The list of reagents used in this study is found in "Reagents and Tools Table".

## *Drosophila* lines and maintenance

Fly stocks and genetic crosses were raised in standard fly media (cornmeal/agar/molasses/yeast) at 18 °C or 25 °C, with 60% humidity and 12 h/12 h dark light cycle, unless stated otherwise. The details of the fly lines used throughout this study are listed in Table EV1. A detailed list of the fly genotypes for each experiment can be found in Table EV2. We used *traffic jam-Gal4* to drive the expression of UAS constructs in the *Drosophila* follicular epithelium.

## In vivo genetic modifier screen

For the genetic modifier screen, the following fly stock was generated: *tj-Gal4*, UAS-Anillin RNAi/CyO; UAS-Myr:GFP, tub-Gal80$^{ts}$/TM6. The RNAi lines for the analyzed cell adhesion molecules were obtained from *Drosophila* Bloomington Stock Center (DBSC), and are listed in detail in Dataset EV1. To fully suppress premature UAS-driven RNAi expression, we used Gal80$^{ts}$, the temperature-sensitive repressor of Gal4, and kept the crosses at 18 °C. In all, 0–4 days after hatching, adult offspring was transferred to 29 °C for 3 days, to boost the efficiency of RNAi depletion. After these 3 days, ovaries were dissected and fixed. At least three fully independent experiments were performed for each RNAi line used and, for each of these experiments, a minimum of 10 egg chambers (randomly selected from a pool of 6–8 dissected animals) were analyzed.

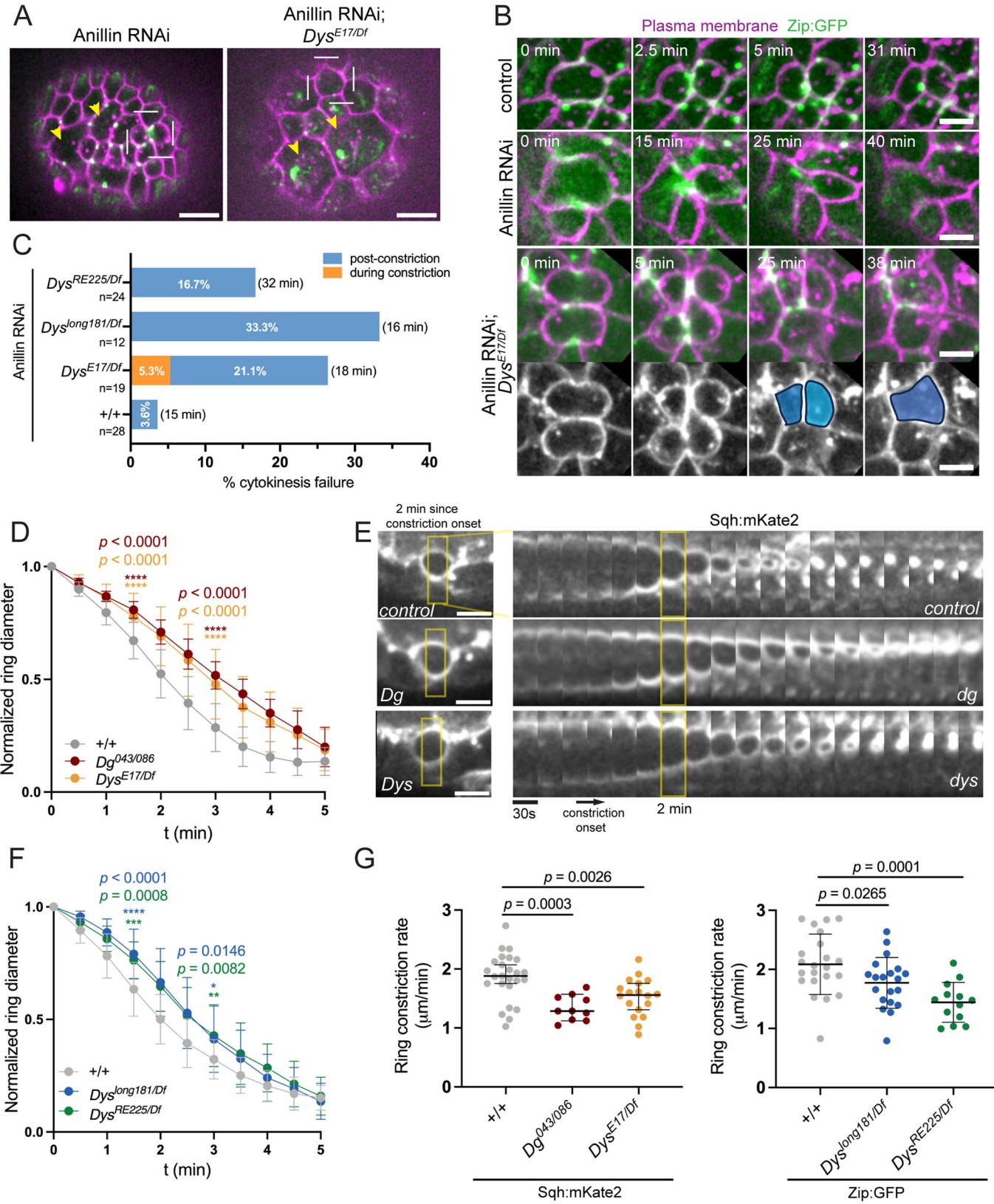

**Figure 5. Roles of the DAPC in epithelial cytokinesis efficiency.**

(A) Surface view of Anillin RNAi and $Dys^{E17/Df}$ Anillin RNAi egg chambers expressing Zip:GFP (green) with stained membranes (CellMask, magenta). Note large cells produced by cytokinesis failure (yellow arrows). Live cytokinesis failure was quantified (B, C) only in cells without previous cytokinesis failure (white box). Scale bars: 10 μm. (B) Time-lapse images (surface view) of Zip:GFP egg chambers with stained membranes captured from control, Anillin RNAi and $Dys^{E17/Df}$, Anillin RNAi. Dys disruption in Anillin-depleted follicle cells leads to cytokinesis failure due to membrane regression after ring closure. Scale bars: 5 μm. (C) Frequency of cytokinesis failure in Anillin-depleted egg chambers on its own ($+/+$) or with concomitant Dys disruption (allelic combinations indicated). For post-constriction failure, average time elapsed from ring constriction end until membrane regression is shown. (D–G) Contractile ring diameter was measured during cytokinesis in control ($n = 25$), $Dg^{043/086}$ ($n = 10$), and $Dys^{E17/Df}$ ($n = 18$) egg chambers expressing Sqh:mKate2 (D, E, G, left) and in control ($n = 24$), $Dys^{long181/Df}$ ($n = 20$), and $Dys^{RE225/Df}$ ($n = 13$) egg chambers expressing Zip:GFP (F, G, right). (E) Stills show 2 min since constriction onset (left) and a close-up (region defined within the yellow box) of cytokinesis progression in control, $Dg^{043/086}$ and $Dys^{E17/Df}$ mutant (right) from a time-lapse of ring constriction viewed along the apical-basal axis. Scale bars: 5 μm. Graphs in (D, F) show mean ± SD through time. Ring diameters were normalized to frame before constriction onset ($t = 0$). P values correspond to the difference in ring diameter between $Dg^{043/086}$ (D, red), $Dys^{E17/Df}$ (D, yellow), $Dys^{long181/Df}$ (F, blue) or $Dys^{RE225/Df}$ (F, green) and the respective controls at $t = 1.5$ and $t = 3$ min. Graphs in (G) show ring constriction rate during the constant constriction phase (mean ± SD). P value calculated using non-parametric unpaired Mann–Whitney test. Source data are available online for this figure.

### Drosophila Schneider (S2) cell culture and transfection

*Drosophila* S2 cells were cultured at 25 °C in Schneider's Insect medium (Sigma-Aldrich) supplemented with 10% fetal bovine serum (FBS) (Thermo Fisher). Transient transfection of mCherry:-Tubulin, pH-DysE:GFP or pUAS-GFP:DysH and pActinGAL4 was performed using the Effectene Transfection Reagent (QIAGEN), according to the manufacturer's instructions. After transfections, cells were incubated at 25 °C for at least 3 days and maximum 5 days prior to the induction of gene expression. In total, $5 \times 10^5$ cells were plated in MatTek culture disks (MatTek; No 1.5; P35G-1.5-7-C) previously coated with poly-D-lysine (Sigma-Aldrich) before performing live imaging.

### Molecular biology and transgenesis

Isoform-specific mutants were generated by CRISPR. gRNAs GGAGGAGCTGAATCTGCAGG and GTGGGAGCTGCTCCTCC-GACG, targeting long (RA, C, F, G, H, I, and K) and RE isoforms, respectively, were cloned in pCFD6 vector (Port and Bullock, 2016) and transgenic lines were generated at attP40 landing site. Then, these lines were crossed with a line expressing Cas9 in the germline and with or without the sfGFP KI at Dys locus. Indel mutations were isolated in the progeny by sequencing. Alleles selected for this work were $Dys^{long181}$ (4 bp deletion, STOP at amino acid position 181 on RH), $Dys^{RE225}$ (2 bp deletion, STOP at amino acid position 225 on RE), $Dys^{long181}$:sfGFP (4 bp deletion, STOP at amino acid position 181 on RH) and $Dys^{RE225}$:sfGFP (deletion of 2 bp, STOP at amino acid position 225).

Dys isoform E cDNA sequence was amplified by PCR from clone RE11449 (*Drosophila* GOLD cDNA collection) and cloned into pENTR. The Gateway cloning system was then used to create pH-DysE:GFP by recombining pENTR-DysE into pHWG through LR Clonase II (Thermo Fisher) mediated recombination. In frame cloning of the pUAS-$Dys^{short}$ transgenes was performed using NEB Builder HiFi DNA Assembly Cloning Kit (New England Biolabs) in pUASz vector (DeLuca and Spradling, 2018), in which EGFP in C-terminal position was previously added. $Dys^{short}$, $Dys^{shortΔSR24}$, $Dys^{shortΔSD}$, $Dys^{shortΔCT}$ encodes amino acids 1–1051, 371–1051, 1–956, 1–714 of the RE isoform, respectively. Transgenes were generated at attP2 site. To generate pUAS-GFP:DysRH, Dys-RH cDNA was synthetized in vitro (Integrated DNA Technologies) in several fragments, before to be cloned in pUASz:GFP vector using NEB Builder HiFi DNA.

### Fixation and staining of egg chambers

*Drosophila* ovaries were dissected in Schneider's Insect Medium (Sigma-Aldrich) supplemented with 10% FBS (fetal bovine serum, heat-inactivated; Thermo Fisher) and fixed using a 4% paraformaldehyde solution (prepared in PBS with 0.05% Tween 20 (Sigma-Aldrich)) for 20 min. After washing three times for 10 min with PBT, samples were mounted with Vectashield Mounting Medium with DAPI (Vector Laboratories). Alternatively, for antibody staining, after the post-fixation washes, egg chambers were blocked for 1 h at room temperature with 10% BSA prepared in PBT and incubated overnight at room temperature with the primary antibody diluted in PBT + 1% BSA. Samples were then washed with PBT + 1% BSA and incubated again for at least two hours at room temperature with the secondary antibody diluted in PBT + 0.1% BSA. After three washing steps with PBT, samples were mounted with Vectashield with DAPI (Vector Laboratories). For F-actin staining, we added Phalloidin-TRITC (Merck, 1:250) to the fixative solution and increased the incubation time to 30 min. The following primary antibodies were used: rat anti-N-cadherin (DSHB DN-Ex #8, 1:50), mouse anti-Fas2 (DSHB 1D4, 1:50), mouse anti-Fas3 (DSHB 7G10, 1:100), mouse anti-βPS-integrin (DSHB CF.6G11-s, 1:10), rabbit anti-Perlecan ((Diaz-Torres et al, 2021), gift from González-Reyes, 1:850) and mouse anti-Armadillo (DSHB N2.7A1, 1:100). Respectively, the following secondary antibodies were used: goat anti-rat Alexa 568 (Invitrogen A11077, 1:300) for N-cadherin, goat anti-mouse Alexa 488 (Invitrogen A11029, 1:300) for Fas2, Fas3 and βPS-integrin, goat anti-rabbit Alexa 488 (Invitrogen A11008, 1:300) for Perlecan and goat anti-mouse Alexa 568 (Invitrogen A11031, 1:300) for Armadillo.

### Imaging

Images of fixed *Drosophila* egg chambers were acquired on an inverted laser scanning confocal microscope Leica TCS SP5 II (Leica Microsystems), with HC PL APO CS 20×/0.70 NA water, 40×/1.10 NA water or 63×/1.30 NA glycerine objectives, using the LAS 2.6 software. For live imaging of *Drosophila* egg chambers, individual ovarioles were dissected in ex vivo culture medium (Schneider's medium (Sigma-Aldrich) supplemented with 10% FBS (fetal bovine serum, heat-inactivated; Thermo Fisher) and 200 μg/μL insulin (Sigma-Aldrich)) and the enveloping muscle removed. Ovarioles were transferred to new culture medium and imaged on glass bottom dishes (MatTek; No 1.5; P35G-1.5-7-C) with an

Andor XD Revolution Spinning Disk Confocal system equipped with two solid state lasers—488 nm and 561 nm—, an iXonEM+ DU-897 EMCCD camera and a Yokogawa CSU-22 unit built on an inverted Olympus IX81 microscope with a PLAPON 60x/1.42 NA objective, using iQ software (Andor). When indicated in the figures, to mark the cell membrane, ovarioles were stained with CellMask Orange Plasma membrane Stain (Thermo Fisher; diluted 1:10,000 in culture medium) for 15 min and washed with ex vivo culture medium before imaging. Live imaging was performed at 25 °C, with the exception for the experiments where UAS-Anillin RNAi was expressed, which were performed at 29 °C. Midsagittal egg chamber cross-sections were used to image the follicular epithelium along the apical-basal axis and z-stacks at the surface of the egg chamber to cross-section the follicular epithelium along the apical-basal axis. Z-stacks were collected with serial optical sections separated by 1 μm. Live imaging of Drosophila S2 cells was performed with the same Andor XD Revolution Spinning Disk Confocal system, using with a UPLSAPO 100×/NA 1.40 objective.

Live imaging of *Drosophila* pupa was conducted as described in (Bosveld et al, 2012), during the first round of cell division in the anterior scutum region of the notum epithelium. Pupa imaging was performed using an inverted spinning disk wide homogenizer confocal microscope (CSU-W1, Roper/Zeiss) equipped with a sCMOS camera (Orca Flash4, Hamamatsu) and using a 63×/1.4 NA oil DICII PL APO objective. A 40 slices Z-stack was collected with serial optical sections separated by 0.5 μm, and captured every 30 s. To obtain apical-basal side views of cytokinesis, a maximum projection was generated from a 2-μm resliced region, centered at the cytokinetic ring.

## Protein extracts and western blot

We prepared protein extracts from *Drosophila* ovaries (at least 25 flies per genotype) of endogenously-GFP-tagged $Dys^{long}$ and $Dys^{short}$ mutants, as well as Dys:sfGFP. Protein extracts of endogenously GFP-tagged aPKC were used as a control to detect an unrelated GFP-tagged protein. Dissected ovaries were transferred to lysis buffer (150 mM KCl, 75 mM HEPES pH 7.5, 1.5 mM EGTA, 1.5 mM MgCl₂, 15% glycerol, 0.1% NP-40, 1× protease inhibitors cocktail (Roche) and 1× phosphatase inhibitors cocktail 3 (Sigma-Aldrich)), and disrupted through sonication. Protein extracts were collected from the supernatant after centrifugation. Samples were resolved by SDS-PAGE and transferred to a nitrocellulose membrane using the iBlot Dry Blotting System (Invitrogen), according to the manufacturer's instructions. Transferred proteins were confirmed by Ponceau staining (0.25% Ponceau S in 40% methanol and 15% acetic acid). The membrane was blocked for at least 1 h at room temperature with 5% dry milk prepared in PBT, and subsequently incubated overnight with the primary antibody (rabbit anti-GFP (i3S core facility, 1:1000)) diluted in blocking solution, at 4 °C. The membrane was then washed three times for 10 min with PBT, and incubated for 1 h with the secondary antibody conjugated to HRP (Jackson ImmunoResearch, 1:5000) diluted in blocking solution, at room temperature. After washing the membrane again three times for 10 min with PBT, blots were developed with Clarity Western ECL Substrate (Bio-Rad) and detected on X-ray films (Fuji Medical).

## Data processing and analysis

Image processing and quantifications were done with FIJI (Schindelin et al, 2012). For representative midsagittal images from egg chambers, we used a single optical section or maximum intensity projections of 2–5 planes. Surface or bottom images of egg chambers correspond to maximum intensity projections of the sections encompassing the epithelial region of interest. For live imaging processing of egg chambers, two FIJI features were always applied: the *StackReg* plugin (EPFL; Biomedical Imaging Group), to correct for the egg chamber movement, and the *Gaussian Blur 3D* filter, to remove background noise. For live imaging of the pupal notum, the *Bleach Correction* plugin was applied to correct bleaching.

Statistical analysis and graphs were generated using GraphPad Prism 9 (GraphPad Software, La Jolla, CA, USA).

### Quantification of multinucleation rate for the Anillin RNAi modifier screen

For the in vivo genetic modifier screen, we scored the number of multinucleated follicle cells in surface z-projections of non-proliferative stage 10 egg chambers ($\Delta z = 1\,\mu m$). We selected stage 10 egg chambers to evaluate the multinucleation ratio for two main reasons. First, follicle cells are no longer undergoing mitotic division, which allows us to directly associate the number of nuclei with the presence of a multinucleated cell. Second, follicle cells are larger at these stages, which facilitates automated segmentation of nuclei and cell membranes. To segment epithelial cells, we developed a set of automated macros for FIJI. The first macro (Computer Code EV1) projects a 1 μm thick section from the acquired z-stack centered around the nucleus; since the nucleus position along the z-axis is variable due to egg chamber curvature, the optimal z-planes to project are determined locally based on the closest nucleus. To extract the multinucleation ratio, the second macro (Computer Code EV2) selects a central region of interest (ROI) of the epithelium (to avoid quantification in the borders of egg chambers whose nuclei/membrane could be missed from the projection), segments the cells (marked with Myristoylated-GFP) within this ROI, segments nuclei in the acquired image (marked with DAPI) and then counts the number of nuclei within the previously segmented cells (see Fig. EV1 for further details). Results from individual images are then summarized in a results table (Computer Code EV3). Manual validation of the segmentation was performed for each image to correct any errors prior to the quantification of the nuclei/cell ratio (multinucleation ratio). In the context of the modifier screen (Fig. 1D–G), the multinucleation ratio for each egg chamber was compared to the mean of a control group of cytokinesis-sensitized (Anillin RNAi) egg chambers expressing UAS-mCherry. This control group was incubated during the same period of time of each of the three independent experiments performed for each RNAi line. Δmultinucleation ratio for each egg chamber is calculated as $(nuclei/cell)_{cell\ adhesion\ RNAi} - (1/n)\,\Sigma(nuclei/cell)i_{control}$, where $n$ represents the number of control egg chambers within each replicate.

### Quantification of multinucleation frequency for interaction experiments with Dys mutants and Tum RNAi

Due to genetic constraints, we used a different genetically encoded marker to visualize the epithelial cell cortex (ECad:GFP) than the one used in the in vivo genetic modifier screen. The number of mono and multinucleated cells was quantified in cross-sections in non-proliferative stage 10 egg chambers. Similar to the approach used for the screen, this quantification was restricted to the central area of the follicular epithelium. Due to the strong multinucleation phenotype produced by Tum RNAi, we used a more restricted

period of incubation at 29 °C (2 days prior to dissection) than the one used for experiments with Anillin RNAi.

### Quantification of ring constriction rate

Analysis of cytokinetic ring constriction was performed in egg chambers expressing a tagged version of non-muscle myosin light chain, Sqh:mKate2 (Fig. 5D,E,G), or non-muscle myosin heavy chain, Zip:GFP (Fig. 5F,G). The diameter of the contractile ring along the apical-basal axis of the epithelium was manually measured using FIJI, from cytokinesis onset ($t_0$) to complete constriction ($t_f$). To depict the change in ring diameter through time (Fig. 5D,F), these values were then normalized to the length of the ring at $t_0$ (defined as the frame prior to a change in ring diameter) and plotted as a function of time, using Prism (GraphPad Software). To measure constriction rate during the constant phase of constriction (~80–40% of the original diameter), we selected four consecutive timepoints for which the change in absolute ring diameter could fit to a linear regression ($R^2 > 0.95$ was set as the threshold for the linear fit to assume constant constriction rate). We determined the constriction rate as the slope (β) of a linear regression (d = α + βt computed with Prism (GraphPad Software)).

### Quantification of cytokinesis failure in time-lapse movies

To understand when epithelial cells failed cytokinesis upon Anillin depletion and/or *Dys* loss-of-function, time-lapse movies were generated in egg chambers expressing Zip:GFP and labeled with a plasma membrane marker. We limited the analysis to small cells to prevent misleading results from defects accumulated from prior cytokinesis failure.

Moreover, we only included in the analysis cells that were imaged at least 20 min after ring constriction. Cells were scored in three categories: (1) not failing cytokinesis, (2) failing cytokinesis during ring constriction, or (3) failing cytokinesis post-constriction.

### Quantification of RFP:Anillin fluorescence intensity

Z-stacks at the surface of control, *Dys^{E17/Df}* and *Dys^{MI025024/Df}* egg chambers were acquired by live imaging to quantify mRFP:Anillin fluorescence intensity during cell division. Z-sum projections encompassing 2 planes separated by 1 μm that cross-sectioned the mitotic cell were used for manual segmentation and quantification of mean pixel intensity at two timepoints: 2 min before the beginning of ring constriction (metaphase) and cytokinesis (2 min after initiation of ring constriction). In metaphase, mRFP:Anillin signal was quantified in the whole cell cortex segmented with a circular ROI (3 pixel width). During cytokinesis, mRFP:Anillin signal was quantified at the cytokinetic ring using a segmented line with ~10 pixel in length and 3 pixel width. The mean cytoplasmic signal measured with a circular ROI with 10 pixel diameter was used for background subtraction of the respective mean pixel intensity of metaphase or cytokinesis signals.

## Data availability

This study includes no data deposited in external repositories.

The source data of this paper are collected in the following database record: biostudies:S-SCDT-10_1038-S44319-024-00319-y.

## Peer review information

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

## Acknowledgements

The authors thank M-L Parmentier, the Bloomington *Drosophila* Stock Center, the Kyoto *Drosophila* Stock Center and the Vienna *Drosophila* Resource center for fly stocks, and C Sunkel for support with supervision. Work in the EM lab is funded by National Funds through FCT—Fundação para a Ciência e a Tecnologia, IP, under the projects UIDB/04293/2020 and PTDC/BIA-CEL/1511/2021, EM is funded by "FCT Scientific Employment Stimulus - Individual Call" program (CEECIND/00622/2017). MG was supported by a PhD fellowship from FCT (SFRH/BD/130708/2017) and by IPATIMUP through CANCER_CHALLENGE2022, which also supported CL. The authors also acknowledge the support of the i3S Scientific Platform ALM, a member of the national infrastructure PPBI - Portuguese Platform of Bioimaging (PPBI-POCI-01-0145-FEDER-022122) and the Institut Curie PICT-IBiSA@BDD imaging facility (member of the French National Research Infrastructure France-BioImaging, ANR-10-INBS-04). Work in the VM lab was supported by the Association Française contre les Myopathies (AFM-Téléthon) (MyoNeurAlp2 network) and by the French government IDEX-ISITE initiative 16-IDEX-0001 (CAP 20-25). Work in the YB lab is supported by the Institut Curie, the CNRS, the INSERM as well as ARC (SL220130607097), ANR (TiMecaDiv 20CE13000801), CANCERO-INCA (PLBIO2020/BELLAICHE) grants.

## Author contributions

**Margarida Gonçalves**: Conceptualization; Formal analysis; Validation; Investigation; Visualization; Methodology; Writing—original draft; Writing—review and editing. **Catarina Lopes**: Formal analysis; Validation; Investigation; Visualization; Writing—review and editing. **Hervé Alégot**: Resources; Methodology. **Mariana Osswald**: Software; Investigation; Writing—review and editing. **Floris Bosveld**: Formal analysis; Investigation; Writing—review and editing. **Carolina Ramos**: Investigation. **Graziella Richard**: Resources; Methodology. **Yohanns Bellaiche**: Supervision; Funding acquisition; Writing—review and editing. **Vincent Mirouse**: Resources; Supervision; Funding acquisition; Writing—review and editing. **Eurico Morais-de-Sá**: Conceptualization; Formal analysis; Supervision; Funding acquisition; Investigation; Visualization; Writing—original draft; Project administration; Writing—review and editing.

Source data underlying figure panels in this paper may have individual authorship assigned. Where available, figure panel/source data authorship is listed in the following database record: biostudies:S-SCDT-10_1038-S44319-024-00319-y.

## Disclosure and competing interests statement

The authors declare no competing interests.

# Expanded View Figures

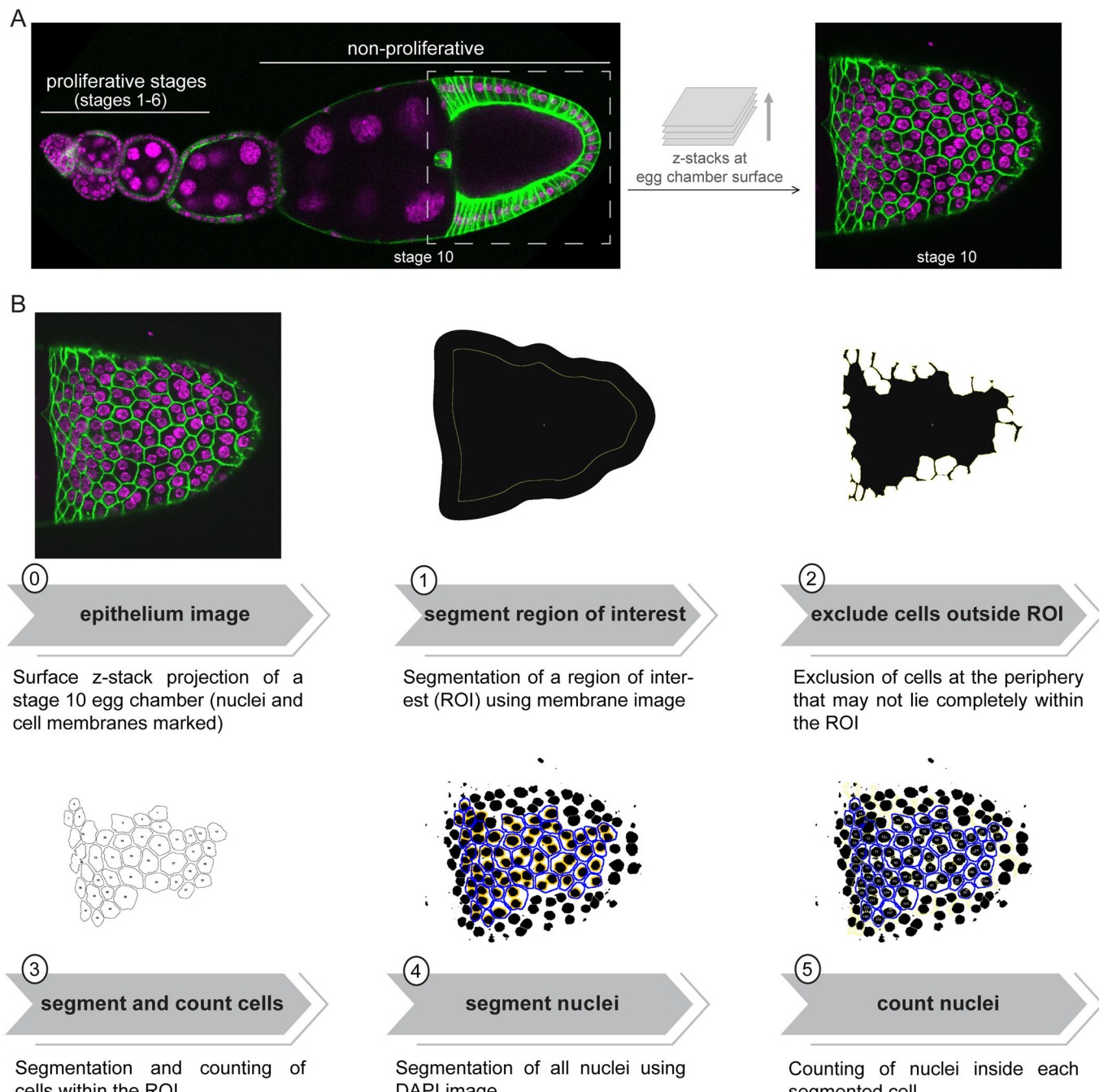

**Figure EV1.  Pipeline for the quantification of cytokinesis defects in the follicular epithelium of stage 10 egg chambers.**

(**A**) Surface z-projections of non-proliferative stage 10 egg chambers were used to quantify the cytokinesis defects from the in vivo genetic modifier screen (nuclei marked with DAPI (magenta) and cell membranes endogenously expressing Myr:GFP (green)). At this stage of oogenesis, follicle cells are no longer undergoing mitosis, allowing us to directly correlate multinucleation with cytokinesis failure, and are larger than at younger stages, facilitating the automated segmentation of nuclei and cell membranes. (**B**) A macro for FIJI was developed for the automated segmentation and counting of nuclei and cells in two-channel images of the follicular epithelium (nuclei in magenta and cell membranes in green). Note that this was restricted to the central area of the egg chamber, to avoid misleading quantifications from nuclei/cells at the periphery of the egg chambers. After running the macro, a manual validation of the segmentation of both nuclei and cells was performed for all images. The obtained values were used to calculate the Multinucleation Ratio, as explained in detail in the Methods section.

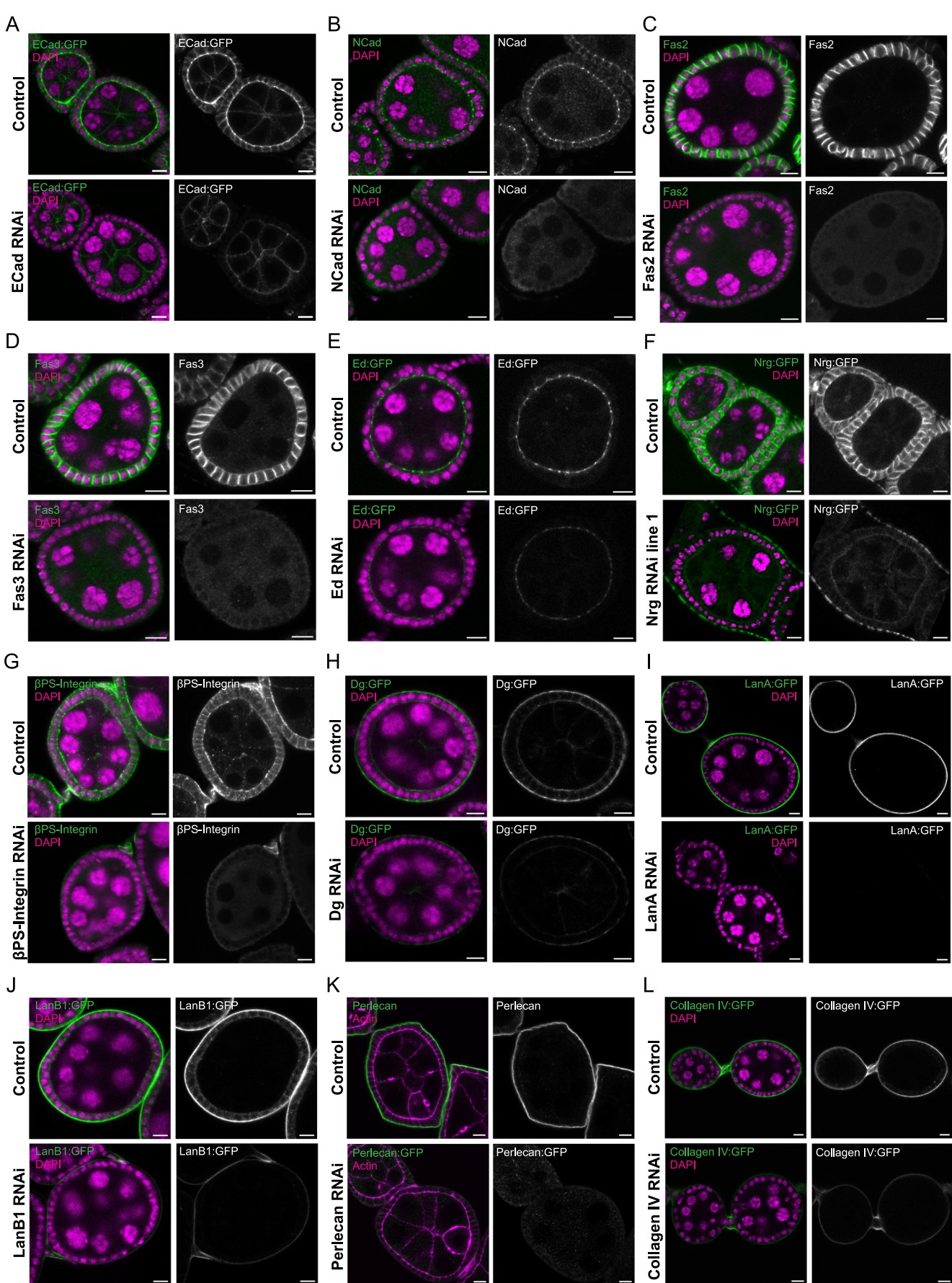

**Figure EV2. Validation of RNAi-mediated depletion of proteins from the genetic modifier screen.**

(A) Midsagittal images of control and ECad RNAi egg chambers endogenously expressing ECad:GFP (green) and DAPI (magenta). (B) Midsagittal images of control and NCad RNAi egg chambers, stained for NCad (green) and DAPI (magenta). (C) Midsagittal images of control and Fas2 RNAi egg chambers, stained for Fas2 (green) and DAPI (magenta). (D) Midsagittal images of control and Fas3 RNAi egg chambers, stained for Fas3 (green) and DAPI (magenta). (E) Midsagittal images of control and Ed RNAi egg chambers endogenously expressing Ed:GFP (green) and stained for DAPI (magenta). (F) Midsagittal images of control and Nrg RNAi (line 1) egg chambers endogenously expressing Nrg:GFP (green) and stained for DAPI (magenta). (G) Midsagittal images of control and βPS-integrin RNAi egg chambers, stained for βPS-integrin (green) and F-actin (magenta). (H) Midsagittal images of control and Dg RNAi egg chambers endogenously expressing Dg:GFP (green) and stained for DAPI (magenta). (I) Midsagittal images of control and LanA RNAi egg chambers endogenously expressing LanA:GFP (green) and stained for DAPI (magenta). (J) Midsagittal images of control and LanB1 RNAi egg chambers endogenously expressing LanB1:GFP (green) and stained for DAPI (magenta). (K) Midsagittal images of control and Perlecan RNAi egg chambers, stained for Perlecan (green) and F-actin (magenta). (L) Midsagittal images of control and Collagen IV RNAi egg chambers endogenously expressing Collagen IV:GFP (green) and stained for DAPI (magenta). (A–L) Scale bars: 10 µm.

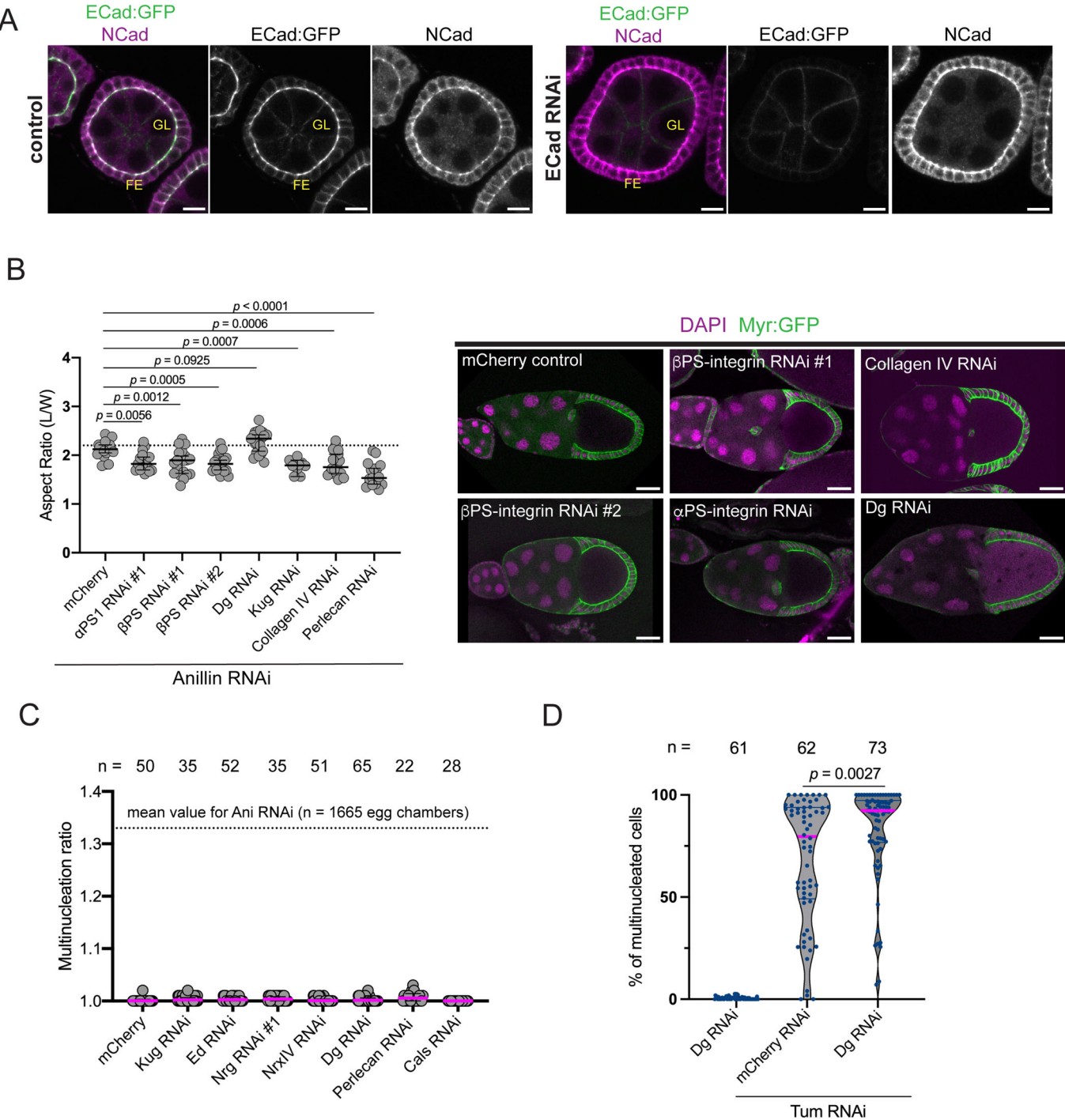

◀ **Figure EV3.  Complementary analysis of RNAi-mediated depletion and their effects on cytokinesis.**

(A) Midsagittal images of control and ECad RNAi egg chambers endogenously expressing ECad:GFP (green) and stained for NCad (magenta). Expression of RNAi for ECad in the follicular epithelium (FE) induces efficient protein depletion. As anticipated, reduction of ECad:GFP fluorescence by ECad RNAi is restricted to the follicular epithelium (FE) and ECad levels are normal in the germline (GL). Scale bar: 10 µm. (B) Midsagittal images and measurement of the aspect ratio in stage 10 egg chambers co-depleted for Anillin and the indicated basal proteins, in comparison with control (UAS-mCherry). Aspect ratio was calculated as the ratio between egg chamber length and width, as depicted in the bottom image (DAPI in magenta and cell membrane in green (Myr:GFP)). Each dot represents the aspect ratio of an analyzed egg chamber. The dashed line represents the anticipated aspect ratio value for this egg chamber stage, as calculated in (Jia et al, 2016). Median ± 95 Confidence Interval is shown. *P* value was calculated by an ordinary one-way ANOVA. Scale bar: 50 µm. (C) Multinucleated ratio in egg chambers depleted for the regulators of cell–cell and cell–matrix interactions that enhanced multinucleation in the Anillin RNAi modifier screen (Fig. 1D–F), in comparison with control (UAS-mCherry). Depletion of these molecules on their own (in an unperturbed cytokinetic background) does not cause multinucleation. Each dot represents the Multinucleation Ratio of an analyzed egg chamber. Median is indicated. The dashed line represents the mean value of the multinucleation ratio for co-expression of Anillin RNAi with the control line (UAS-mCherry). (D) Frequency of multinucleated cells in egg chambers either co-expressing UAS-Dg RNAi with UAS-mCherry RNAi (UAS titulation control), UAS-Tum RNAi with UAS-mCherry RNAi or UAS-Tum RNAi with UAS-Dg RNAi. UAS-driven depletion of Tum causes strong multinucleation frequency in the follicular epithelium, which is significantly increased by the concomitant depletion of Dg. Violin plots and median (magenta) is indicated. Each dot represents an egg chamber. Sample size (n) is indicated on top. *P* value calculated by non-parametric unpaired Mann–Whitney test.

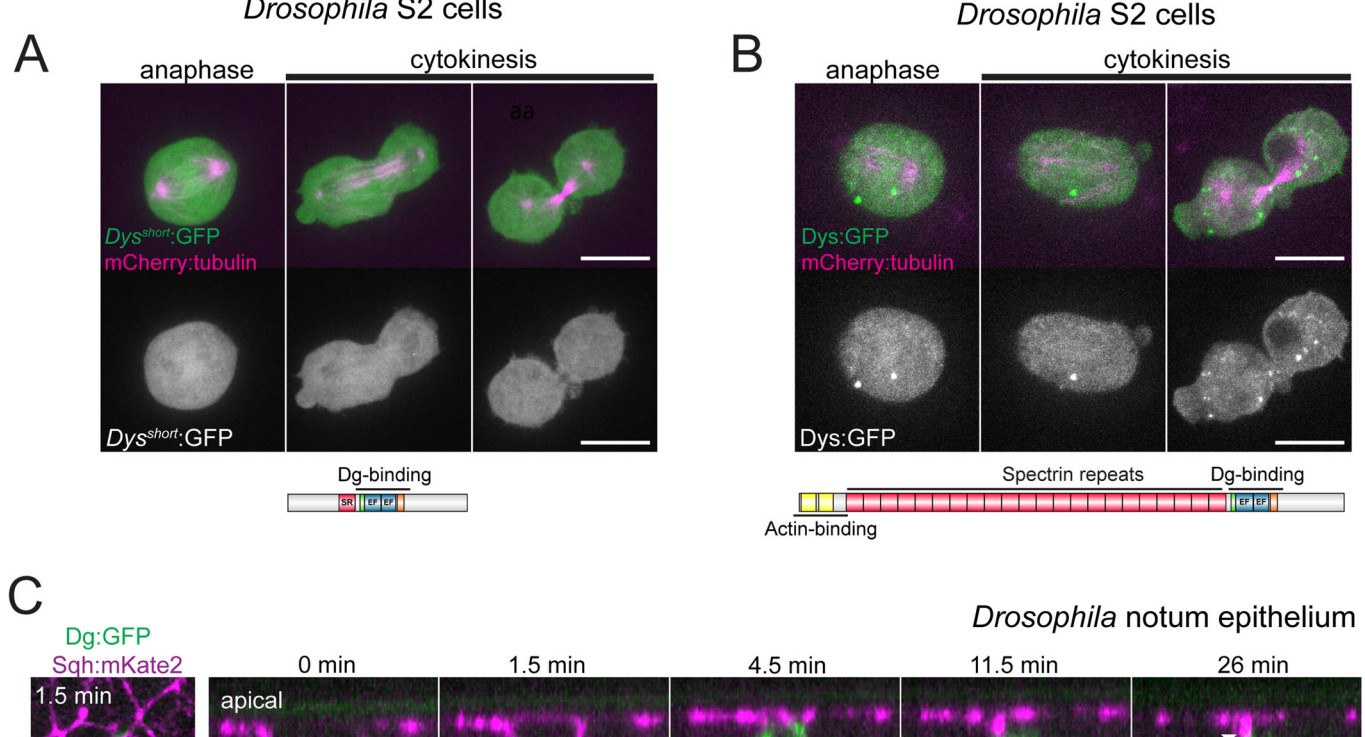

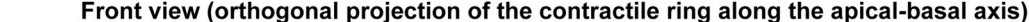

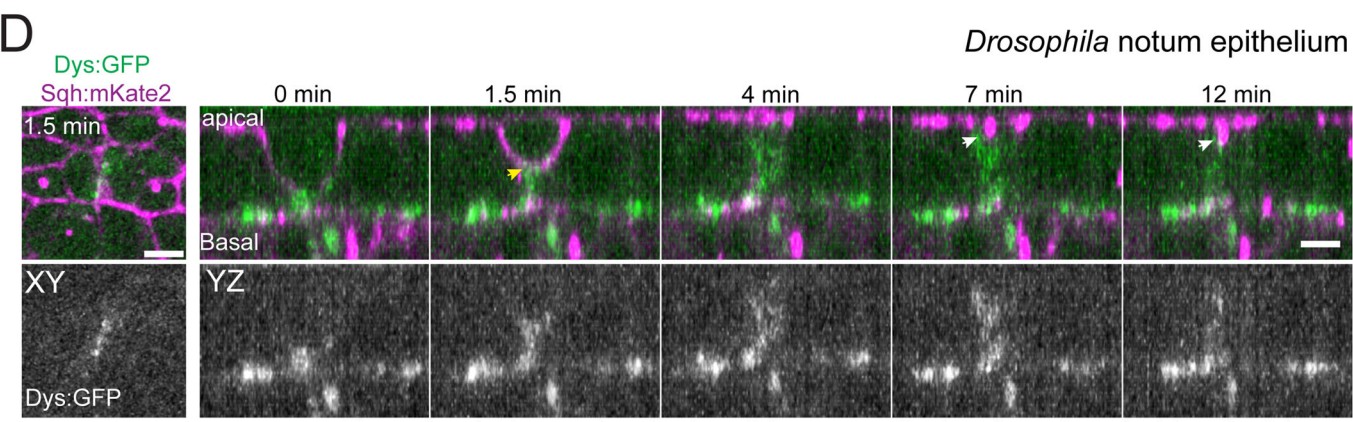

**Figure EV4.   Dynamic redistribution of the DAPC during epithelial cytokinesis in *Drosophila* S2 cells and pupal notum epithelium.**

(**A, B**). The short Dys isoform E (**A**) and the long isoform H (**B**) do not show any local accumulation during cytokinesis in *Drosophila* S2 cells. Dys isoforms are GFP-tagged versions (green) and mCherry:Tubulin labels microtubules (magenta). Scale bars: 10 μm. (**C, D**). Redistribution of endogenously tagged Dg:GFP (**A**) and Dys:GFP (**B**) during epithelial cytokinesis in the *Drosophila* pupal dorsal thorax (notum). During ring (labeled with Sqh:mKate2) constriction, Dg:GFP (**A**) and Dys:GFP (**B**) accumulate at the basal part of the ingressing membrane (yellow arrows). After ring closure, both proteins become enriched close to the midbody (white arrows). Scale bars: 5 μm.

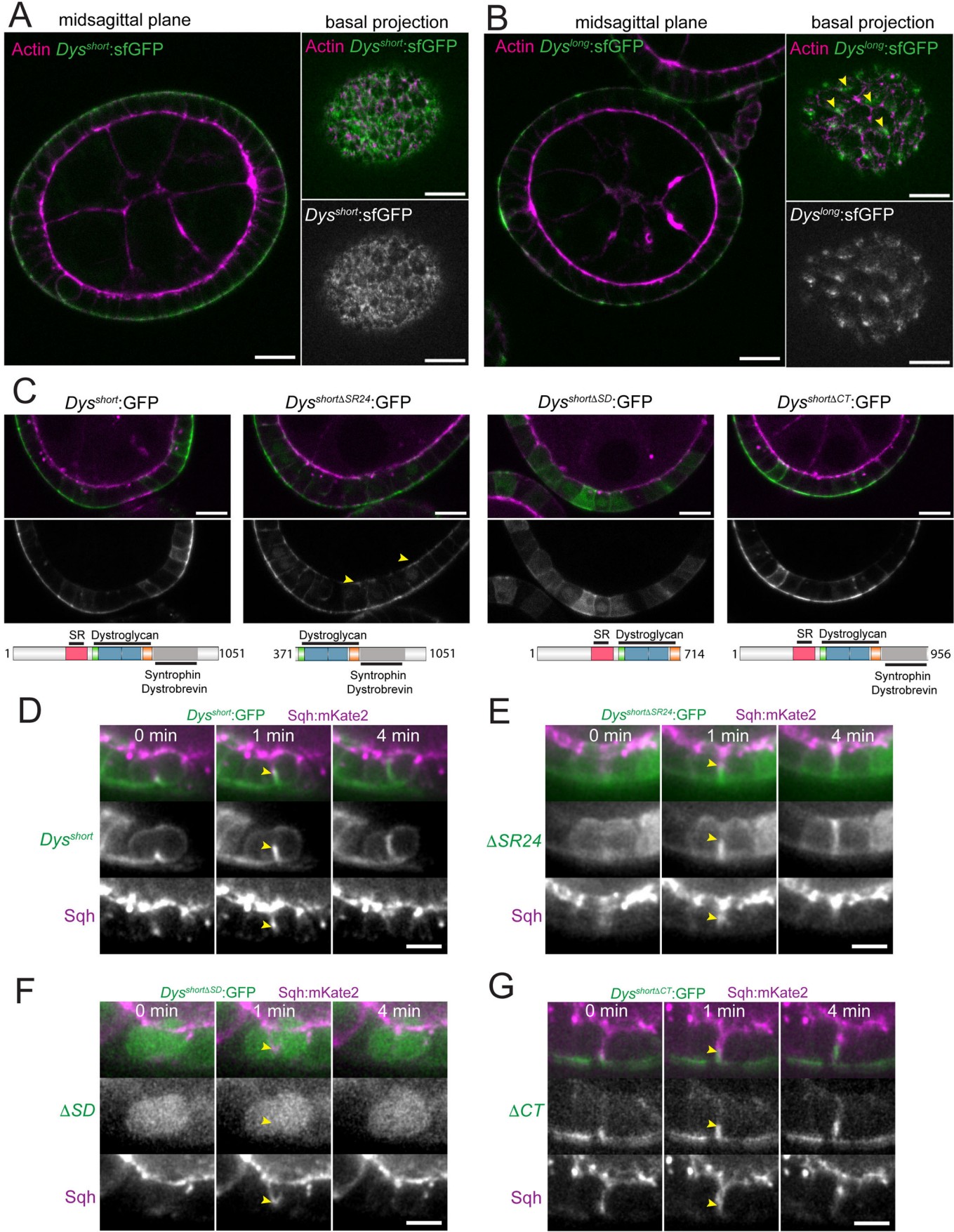

**Figure EV5.   Spatial distribution of Dystrophin isoforms and truncated versions.**

(A) Midsagittal images (left) and basal projection (right) of the proliferative follicular epithelium endogenously expressing *Dys^short*:sfGFP (green) and stained for actin (magenta). *Dys^short* mainly localizes at the basal domain of epithelial cells. Scale bar: 10 μm. (B) Midsagittal images (left) and basal projection (right) of the proliferative follicular epithelium endogenously expressing *Dys^long*:sfGFP (green) and stained for actin (magenta). *Dys^long* mainly localizes to the basal domain in a planar polarized manner. Scale bar: 10 μm. (C) Midsagittal images of egg chambers expressing UAS-driven *Dys^short*:GFP, *Dys^{shortΔSR24}*:GFP, *Dys^{shortΔSD}*:GFP or *Dys^{shortΔCT}*:GFP and expressing Sqh:mKate2 (magenta). The short *Dys* isoform mainly localizes at the basal domain of epithelial cells, which is not altered by the deletion of the most downstream C-term region (*Dys^{shortΔCT}*). Partial mislocalization to the apical domain (yellow arrows) is observed in egg chambers expressing *Dys^{shortΔSR24}*:GFP. Deletion of syntrophins/Dystrobrevin interaction sites leads to Dys mislocalization and only weak basal enrichment (*Dys^{shortΔSD}*). Note that these UAS-driven constructs show mosaic expression levels in the follicular epithelium. Scale bar: 10 μm. (D–G) Time-lapse images of follicle cells expressing UAS-driven *Dys^short*:GFP, *Dys^{shortΔSR24}*:GFP, *Dys^{shortΔSD}*:GFP or *Dys^{shortΔCT}*:GFP. *Dys^short* (A) becomes enriched at the ingressing membrane during ring constriction, which is labeled with Sqh:mKate2 (position of the basal part of the ring is marked with arrows). *Dys^{shortΔSR24}* (E) and *Dys^{shortΔCT}* (G) present a similar cytokinetic redistribution. In contrast, removal of the syntrophins/Dystrobrevin binding sites prevents Dys accumulation in the ingressing membrane during epithelial cytokinesis (F). Scale bar: 5 μm.

