## [Peer Review File · EMBO Reports]

The Dystrophin-Dystroglycan complex ensures cytokinesis efficiency in *Drosophila* epithelia

Margarida Gonçalves, Catarina Lopes, Hervé Alégot, Mariana Osswald, Floris Bosveld, Carolina Ramos, Graziella Richard, Yohanns Bellaiche, Vincent Mirouse, and Eurico Morais-de-Sá

Corresponding author(s): Eurico Morais-de-Sá (eurico.sa@ibmc.up.pt)

Review Timeline:

Submission Date:	4th Mar 24
Editorial Decision:	19th Apr 24
Revision Received:	25th Aug 24
Editorial Decision:	16th Oct 24
Revision Received:	21st Oct 24
Accepted:	24th Oct 24

Editor: Deniz Senyilmaz Tiebe

Transaction Report:

Dear Eurico,

Thank you for the submission of your research manuscript to our journal, which was now seen by three referees, whose reports are copied below.

My apologies for this unusual delay in getting back to you. It took longer than anticipated to receive the full set of referee reports.

Referees express interest in the proposed role of the Dystrophin-Dystroglycan complex in cytokinesis. However, they also raise significant concerns that need to be addressed to consider publication here.

I find the reports informed and constructive, and believe that addressing the concerns raised will significantly strengthen the manuscript. As the reports are below, and I think all points need to be addressed, I will not detail them here.

Given these positive recommendations, we would like to invite you to submit a revised manuscript. Please revise your manuscript with the understanding that the referee concerns (as in their reports) must be fully addressed and their suggestions taken on board. Please address all referee concerns in a complete point-by-point response. Acceptance of the manuscript will depend on a positive outcome of a second round of review. It is EMBO reports policy to allow a single round of major experimental revision only and acceptance or rejection of the manuscript will therefore depend on the completeness of your responses included in the next, final version of the manuscript.

We realize that it is difficult to revise to a specific deadline. In the interest of protecting the conceptual advance provided by the work, we recommend a revision within 3 months. Please discuss the revision progress ahead of this time with me if you require more time to complete the revisions, or if you have questions or comments regarding the revision (also by video chat).

1. A data availability section providing access to data deposited in public databases is missing (where applicable).
2. Your manuscript contains statistics and error bars based on $n=2$. Please use scatter plots in these cases.

You can submit the revision either as a Scientific Report or as a Research Article. For Scientific Reports, the revised manuscript can contain up to 5 main figures and 5 Expanded View figures, and it should not exceed 27000 characters. If the revision leads to a manuscript with more than 5 main figures it will be published as a Research Article. In this case the Results and Discussion section should be separate. If a Scientific Report is submitted, these sections have to be combined. This will help to shorten the manuscript text by eliminating some redundancy that is inevitable when discussing the same experiments twice. In either case, all materials and methods should be included in the main manuscript file.

3) We replaced Supplementary Information with Expanded View (EV) Figures and Tables that are collapsible/expandable online. A maximum of 5 EV Figures can be typeset. EV Figures should be cited as 'Figure EV1, Figure EV2' etc... in the text and their respective legends should be included in the main text after the legends of regular figures.

4) a .docx formatted letter INCLUDING the reviewers' reports and your detailed point-by-point responses to their comments. As part of the EMBO publication's Transparent Editorial Process, EMBO reports publishes online a Review Process File (RPF) to accompany accepted manuscripts. This File will be published in conjunction with your paper and will include the referee reports,

your point-by-point response and all pertinent correspondence relating to the manuscript.

<https://www.embopress.org/page/journal/14693178/authorguide#transparentprocess>

5) a complete author checklist, which you can download from our author guidelines

<https://www.embopress.org/page/journal/14693178/authorguide>. Please insert information in the checklist that is also reflected in the manuscript. The completed author checklist will also be part of the RPF.

6) Please note that all corresponding authors are required to supply an ORCID ID for their name upon submission of a revised manuscript (). Please find instructions on how to link your ORCID ID to your account in our manuscript tracking system in our Author guidelines

7) Before submitting your revision, primary datasets produced in this study need to be deposited in an appropriate public database (see <https://www.embopress.org/page/journal/14693178/authorguide#datadeposition>). Please remember to provide a reviewer password if the datasets are not yet public. The accession numbers and database should be listed in a formal "Data Availability" section placed after Materials & Method (see also

<https://www.embopress.org/page/journal/14693178/authorguide#datadeposition>). Please note that the Data Availability Section is restricted to new primary data that are part of this study. * Note - All links should resolve to a page where the data can be accessed. *

Additional information on source data and instruction on how to label the files are available:

<https://www.embopress.org/page/journal/14693178/authorguide#sourcedata>

9) Our journal encourages inclusion of *data citations in the reference list* to directly cite datasets that were re-used and obtained from public databases. Data citations in the article text are distinct from normal bibliographical citations and should directly link to the database records from which the data can be accessed. In the main text, data citations are formatted as follows: "Data ref: Smith et al, 2001" or "Data ref: NCBI Sequence Read Archive PRJNA342805, 2017". In the Reference list, data citations must be labeled with "[DATASET]". A data reference must provide the database name, accession number/identifiers and a resolvable link to the landing page from which the data can be accessed at the end of the reference. Further instructions are available at <http://www.embopress.org/page/journal/14693178/authorguide#referencesformat>

10) Regarding data quantification (see Figure Legends:

<https://www.embopress.org/page/journal/14693178/authorguide#figureformat>)

11) The journal requires a statement specifying whether or not authors have competing interests (defined as all potential or actual interests that could be perceived to influence the presentation or interpretation of an article). In case of competing interests, this must be specified in your disclosure statement. Further information: <https://www.embopress.org/competing->

interests

12) Please also note our reference format:

I look forward to seeing a revised version of your manuscript when it is ready. Please let me know if you have questions or comments regarding the revision.

Kind regards,

Deniz

Deniz Senyilmaz Tiebe, PhD
Scientific Editor
EMBO Reports

Referee #1:

Goncalves et al., present an interesting manuscript detailing contributions of Dystrophin/Dystroglycan complex to cytokinesis completion in the *D. melanogaster* follicular epithelium. These proteins link the extracellular matrix to the cytoskeleton and this manuscript sheds light on how the connection of adhesion receptors and matrix interactions can control cytokinesis progression. A real strength of the manuscript is that it investigates cytokinesis in an in-vivo setting and uses some really delicate and state-of-the-art techniques (optogenetics, endogenous tagging, a modifier screen) to investigate this axis. The enhancers and suppressors of cytokinesis identified in the screen will provide many new insights into cytokinesis completion. The phenotypes are generally clear and the interplay of Dystrophin/Dystroglycan and Anillin is interesting. The weaknesses of the manuscript are that it isn't clear how these proteins contribute toward cytokinesis progression, what their role in the absence of Anillin depletion is, why both long and short isoforms are needed for completion of cytokinesis or what these proteins are actually doing to allow cytokinesis to occur. The manuscript is generally well written, I have some suggestions below:

1. The authors are clear that the cytokinesis phenotypes observed upon depletion are only observed in the background of mild Anillin depletion. Whilst they interpret this as phenotypes being exposed in a sensitized background, it may be that that phenotypes relate only to Anillin-depleted state. Can you show that disruption of the Dystrophin/Dystroglycan axis induces cytokinesis failure in other sensitized backgrounds? Can you show whether Anillin itself is further destabilised in the Dystrophin/Dystroglycan depleted state; this would help confirm that the phenotypes are direct and independent of Anillin.
2. I found it quite hard to follow what the authors were claiming in Figure 3 related to the movement of Dg:GFP and Dys:GFP. I appreciate that endogenous labelling is technically challenging, but is there a way to be clearer about what is a contractile ring and what is a midbody? Does the Dys:GFP and Dg:GFP spread from the basolateral domain over the contractile ring?
3. The redistribution of Dysshort:sfGFP and Dyslong:sfGFP to the contractile ring suggests it is independent of actin binding; is this redistribution dependent upon Anillin?
4. The cytokinesis failure in lines just expressing the short or the long isoform of Dystrophin is clear and impaired contractile ring ingression in the absence of dystrophin or dystroglycan is clear, but how are these proteins contributing to these processes - are there any elements within Dys that are needed for cytokinesis?
5. Is the GFP-tagging of Dystrophin functional? Given that neither Dysshort:sfGFP nor Dyslong:sfGFP can support cytokinesis in the absence of Anillin, is it possible that these are functionally compromised fusion proteins?

Minor

L188. A call out to Fig 1F would help

The renaming of the isoform-specific replacements as Dysshort:sfGFP and Dyslong:sfGFP is helpful, but this renaming seems only to have been applied to the main text - including these labels on Figures 3 and 4 markup would help readability.

Referee #2:

Cytokinesis of the epithelial cells is not trivial since the cell-cell adhesion needs to be remodelled without affecting the barrier function of the tissue. In this manuscript, the authors performed an RNAi-based targeted screen for the modifiers of the cytokinesis failure due to the depletion of anillin, a widely conserved multifunctional linker protein crucial for cytokinesis. Cell-adhesion molecules such as cadherins and immunoglobulin-superfamily molecules, extracellular matrix proteins, and their receptors were examined. Among the positive hits, the authors focused on dystrophin-associated protein complex (DAPC). Deficiency of dystrophin (*dys*), light-induced artificial aggregation of *Dys*, and specific depletion of short or long isoform all enhanced the cytokinesis failure by anillin RNAi. Live observation revealed that dystrophin isoforms are enriched to the ingressing membrane during ring constriction and that dystrophin mutants (+anillin RNAi) show late cytokinesis phenotype. Delayed ring constriction was also observed in the dystrophin and dystroglycan mutants.

This work demonstrated that the DAPC contributes to the robustness of cytokinesis in epithelial tissues. It is suitable for publication in EMBO Reports after addressing a few points below.

Major points

1. Specificity of the role of DAPC in the epithelial cytokinesis.

Although the authors performed the screening and further analysis in the epithelial tissues, it is left unclear whether the role of DAPC for robust cytokinesis is specific to the epithelial tissues or not. Testing the enhancement of the anillin phenotype in non-epithelial tissues would strengthen the conclusion. Otherwise, it should be clearly stated that the role of DAPC might not be specific to epithelial cytokinesis.

2. Variation of the control in the screen.

Figs. 1D~G show that the spread of nuclei/cell in the mCherry RNAi + anillin RNAi strain varies from experiment to experiment. Its mean and median are also likely to vary. However, plotting the data by " Δ Multinucleation Ratio" prohibits the readers from checking this. Simply plotting 'nuclei/cell' would be more informative. At least, two columns for the mean and standard deviation of the mCherry control should be added in Table EV1.

3. Timing of furrowing

In Figs 4D~G, no difference was found in the rate of furrow constriction. The observed difference in the timing strongly depends on how the timing of the onset of furrowing ($t=0$) is defined. In the bottom row of Fig. 4E, there is no visible change of the diameter in the first 2 min after "constriction onset". An independent and more reliable standard such as anaphase onset by observing chromosomes is necessary to demonstrate the rather subtle difference.

Minor points

(Fig. 1D~G legend)

Δ Multinucleation Ratio = (nuclei/cell)RNAi - mean (nuclei/cell)mCherry

(line 533 Materials and Methods)

(nuclei/cell)cell adhesion RNAi - $(1/n) * \sum(X(\text{nuclei/cell})_i)$ control, where n represents the number of control egg chambers within each replicate.

What is X()?

Fig. 2F and G

missing control for the non-specific effect of light exposure. For example, 72 h light exposure of the *Dys*:sfGFP VHH:CY2 stain without CIBN:MP should not show cytokinesis failure.

Referee #3:

Cytokinesis, the process of separating daughter cells after cell division, can be challenging for epithelial cells. These cells are connected to neighboring cells and the extracellular matrix through transmembrane protein complexes. To understand how cell adhesion machinery affects cytokinesis efficiency, Gonçalves and cols conducted an RNAi-based study in the *Drosophila* follicular epithelium. They found that certain adhesion molecules and transmembrane receptors aid cytokinesis completion. They show that Dystrophin/Dystroglycan, which link the extracellular matrix to the cell's cytoskeleton, concentrate in the ingressing membrane below the cytokinetic ring during and after constriction. Functional analyses let the authors suggest that the machinery involved in cell-cell and cell-matrix interactions doesn't hinder cytokinesis; instead, it has evolved to ensure efficient cell division in epithelial tissues.

The findings presented here shed light on how cell-cell and cell-matrix interactions contribute to efficient cell division in epithelial tissues. Overall, this work provides valuable insights into cellular processes and has broader implications for understanding tissue development and, potentially, health. This study is well presented, written, and organized, and holds general interest.

I would however suggest addressing some specific points (see below), before the MS is ready for publication:

The authors use a cytokinesis-sensitized background produced by RNAi against Anillin. It would be interesting to test if the

effects observed by Dystrophin/Dystroglycan (over-expression and depletion) is specific for that background or can also modify the effects of other elements involved in cytokinesis other than anillin. These experiments would show if the effect is specific for anillin depletion or has a more general role in the cytokinesis process.

In Fig 1B, C, the authors should show if UAS-cad, on its own, has an effect in cytokinesis, independently of Anillin-RNAi. This control is needed to better interpret their results.

The authors provide some analysis of RNAi efficiency by immunofluorescence, but this is only limited to some of the genes analyzed. It would be important to use those techniques or qPCR in the other conditions analyzed. This would validate the use of those tools and could allow to present that lack or presence of phenotypes with more confidence.

In some panels, the authors use the "UAS" term to label the genotypes (eg, Fig 1C) in others they don't (Fig 1D). All those transgenes are UAS-driven. I would suggest being consistent with the nomenclature. Otherwise, it could be confusing to the reader. For example, in Fig 1F, the authors label a data set as UAS Dg-C, but the other transgenes (eg mCherry, or Dg RNAi) do not have the UAS in them. For consistency, I would suggest reviewing that.

We thank the reviewers for their positive evaluation of our work and constructive comments and suggestions. In response, we have included a number of new experiments that have significantly improved our manuscript, from which we would like to highlight:

- As requested by reviewers 1 and 3, **we have included data further supporting that the role of the Dys/Dg complex in cytokinesis is not solely related to Anillin function.** We show that Dg RNAi also enhances the cytokinesis phenotype produced by depletion of Tum, a component of the centralspindlin complex that acts independently of Anillin in multiple aspects of cytokinesis (**Fig EV3**), and we provide evidence that Anillin recruitment to the cytokinetic ring is unaffected by loss of Dystrophin function (**Fig 2D,2E**).
- **We investigated which common molecular domains shared between Dys isoform may control the cytokinetic function of Dys,** in line with a comment of reviewer 1. Our data show that the cytokinetic distribution of Dys is disrupted by deletion of evolutionary conserved C-terminal motifs that are known to interact with syntrophins and dystrobrevin (**Fig EV5E-EV5G**).
- In response to reviewer 2, **we have imaged the cytokinetic distribution of Dystrophin in two non-epithelial cell types that are classically used to study cell division.** We found that Dystrophin is either not expressed (neuroblasts) or localized uniformly (S2 cells, **Fig EV4A, EV4B**). Nevertheless, the DAPC may have cytokinetic functions beyond epithelia and we refer to this point in the revised manuscript.

We should note that we removed the set of light-induced clustering experiments (**Fig 2E-G of the original manuscript**). Following a control experiment suggested by reviewer 2, we found a non-specific effect produced by temperature increase inside the optogenetic box upon light activation. Since the phenotypic interaction between the DAPC and Anillin is supported by complementary experiments (**Fig 1F, Fig 2B, 2C and Fig 4C, 4D**), and the role of the complex during cytokinesis was addressed by live imaging (**Fig 3, Fig 4B, Fig EV4C, EV4D, Fig 5**), we believe that the main conclusions of the manuscript remain fully supported even without this dataset.

We also include additional data (**Fig 1C, Fig 3E, Fig EV2**) and carefully edited text, figures and tables to address other reviewer's comments and improve the manuscript, **as detailed in the following point-by-point response.**

Reviewer #1:

Goncalves et al., present an interesting manuscript detailing contributions of Dystrophin/Dystroglycan complex to cytokinesis completion in the D. melanogaster follicular epithelium. These proteins link the extracellular matrix to the cytoskeleton and this manuscript sheds light on how the connection of adhesion receptors and matrix interactions can control cytokinesis progression. A real strength of the manuscript is that it investigates cytokinesis in an in-vivo setting and uses some really delicate and state-of-the-art techniques (optogenetics, endogenous tagging, a modifier screen) to investigate this axis. The enhancers and suppressors of cytokinesis identified in the screen will provide many new insights into cytokinesis completion. The phenotypes are generally clear and the interplay of Dystrophin/Dystroglycan and Annilin is interesting. The weaknesses of the manuscript are that it isn't clear how these proteins contribute toward cytokinesis progression, what their role in the absence of Annilin depletion is, why both long and short isoforms are needed for completion of cytokinesis or what these proteins are actually doing to allow cytokinesis to occur. The manuscript is generally well written, I have some suggestions below:

- 1. The authors are clear that the cytokinesis phenotypes observed upon depletion are only observed in the background of mild Annilin depletion. Whilst they interpret this as phenotypes being exposed in a sensitized background, it may be that that phenotypes relate only to Annilin-depleted state. Can you show that disruption of the Dystrophin/Dystroglycan axis induces cytokinesis failure in other sensitized backgrounds?*

Response: We thank the reviewer for this comment (shared with reviewer 3), which prompted us to test whether Dystrophin/Dg function would affect other backgrounds sensitised to cytokinesis failure by tissue-specific depletion of 3 genes involved in different aspects of cytokinesis:

- **Tumbleweed** (RACGAP1 in mammals) is a component of the centralspindlin complex that acts during cytokinesis from ring assembly to midbody formation and abscission (PMID: 22927365). We found that Tum RNAi alone causes dramatic cytokinesis defects in the follicular epithelium. We thereby used the GAL80ts module under 2 days of incubation at 29°C (3 days were used in the screen) to further restrict the period of RNAi expression. Under these conditions, Tum RNAi still produced strong cytokinesis defects (11 % of egg chambers (n = 62) showed all follicle cells multinucleated; mean frequency of multinucleated cells = 68,3 %). Importantly, there is a significant enhancement of multinucleation by co-expression of Dg RNAi (**Fig EV3D**, 23 % of egg chambers (n = 73) showed all follicle cells multinucleated; mean frequency of multinucleated cells = 82,4 %, p = 0,0027).

- **Rok** (ROCK in mammals) is involved in contractile ring formation and constriction (PMID: 16488869). In contrast to Tum RNAi, follicle cell-specific depletion of Rok induced only

very weak multinucleation defects (48% of egg chambers (n = 50) **without** any multinucleated cells; mean frequency of multinucleated cells = 0.82%). This weak phenotype is not ideal for evaluating genetic interactions, as the efficiency of cytokinesis completion may not be properly challenged under these conditions. Nevertheless, we observed a slight increase in multinucleation frequency by co-expressing Dg RNAi (**Fig Rev1**, 19 % of egg chambers (n = 52) **without** any multinucleated cell; mean frequency of multinucleated cells = 1, 34 %, p=0.023). Given the very weak multinucleation phenotype under these conditions, we did not consider this result compelling enough to include in the manuscript.

- The Septin **Peanut** (Septin7 in mammals) is required for cytokinesis by forming Septin filaments that associate with Anillin and help to attach the contractile ring and subsequently the midbody ring to the membrane (PMID: 8181057, PMID: 22804577). Unfortunately, we were unable to test a possible interaction between this Septin (RNAi line: P(TRiP.HMC05924)attP40 - BL accession # 65157) and the DAPC complex (Dg RNAi or Dys mutants), because Pnut depletion caused severe defects in the morphogenesis of egg chambers, which failed to reach stage 10 of oogenesis. For this experiment, we generated flies with the following genotypes: Pnut RNAi (tj-Gal4/UAS-Pnut RNAi; UAS-mCherry RNAi/+) vs. Pnut RNAi +Dg RNAi (tj-Gal4/UAS-Pnut RNAi; UAS-Dg RNAi/+) or Pnut RNAi (tj-Gal4/UASPnut RNAi) vs Pnut RNAi + Dys mutants (tj-Gal4/Pnut RNAi; Dys MI025024/DysDf), and all of them lacked stage 10 egg chambers.

In conclusion, we have included the interaction analysis with Tum in the manuscript to provide evidence that DAPC can promote cytokinesis efficiency under different conditions that sensitize cells to cytokinesis failure. In addition, we note that the manuscript also describes cytokinetic phenotypes observed in DAPC loss of function without perturbation of Anillin, reinforcing that DAPC function is not solely related to the Anillin-depleted state.

Figure Rev1: Assessing the impact of Dg depletion on the frequency of multinucleated cells produced by Rok RNAi Frequency of multinucleated cells in stage 10 egg chambers expressing UAS-driven Rok RNAi alone (genotype: *tj-Gal4, ECadGFP/+; UAS-Rok RNAi (P{TRiP.JF03225})/+*; mean = 0.82) or combined with UAS-Dg RNAi (genotype: *tj-Gal4, ECad-GFP/+; UAS-Rok RNAi/UAS-Dg RNAi*; mean = 1,34) in the follicular epithelium. Each dot represents the frequency of multinucleated cells in an egg chamber. *p*-value calculated by non-parametric unpaired MannWhitney test.

Can you show whether Anillin itself is further destabilised in the Dystrophin/Dystroglycan depleted state; this would help confirm that the phenotypes are direct and independent of Anillin

Response: We tried to evaluate whether Anillin would be further destabilized if we depleted Anillin and Dystroglycan simultaneously. However, the signal of Anillin was barely detectable under Anillin depletion conditions, making it impossible to rigorously evaluate any further changes in Anillin stability (**Fig Rev2**). Therefore we determined whether there was any effect of the DAPC on Anillin distribution by imaging mRFP:Anillin in *Dys^{E17/Df}* and *Dys^{M1025024/Df}* mutant egg chambers. These experiments (**Fig 2D, 2E**), show that Dystrophin is not required to control the cytokinetic accumulation of mRFP-Anillin, suggesting that Dystrophin does not directly contribute to Anillin function. We mention this point in the manuscript “A potential role of Dys on Anillin recruitment to the contractile ring could explain this phenotypic enhancement. However, mRFP:Anillin redistribution during cytokinesis was unaffected by loss of Dys function (Fig 2D, E). We conclude that the DAPC promotes cytokinesis efficiency but does not directly contribute for Anillin cytokinetic recruitment.”

Figure Rev2: RNAi-mediated depletion of Anillin leads to reduced detection of Anillin:GFP in follicle cells. Arrows indicate the position of ring canals where Anillin is strongly enriched during interphase. mCherry

RNAi: *tj-Gal4/+; UAS-Anillin:GFP/UAS-mCherry* RNAi. *Anillin* RNAi: *tj-Gal4 /UAS-Anillin* RNAi; *Anillin:GFP/UAS-mCherry* RNAi. Scale bar: 20 μ m

2. I found it quite hard to follow what the authors were claiming in Figure 3 related to the movement of *Dg:GFP* and *Dys:GFP*. I appreciate that endogenous labelling is technically challenging, but is there a way to be clearer about what is a contractile ring and what is a midbody? Does the *Dys:GFP* and *Dg:GFP* spread from the basolateral domain over the contractile ring?

Response: To address this comment, we have redefined what is a contractile ring and a midbody in the initial part of the text “and mRFP:Anillin, which labels first the cytokinetic ring and then the midbody as it is assembled at the end of constriction”. Moreover, we performed a number of revisions in **Figure 3** to improve the clarity of *Dys/Dg* localization in relation to the contractile ring and midbody. Namely:

- We included separated channels for mRFP:Anillin and Sqh-mKate as both proteins label the contractile ring and subsequently the midbody upon ring closure.
- We included arrows as reference points in these multiple channels, also including labels to indicate ring and midbody positions in **Fig 3A** and **Fig 3D** (kymograph)
- To further clarify the description of localization at the midbody stage, we performed coimaging of *Dys:GFP* and Tubulin-RFP in surface projections to directly label the midbody associated microtubules, confirming that *Dys* is still enriched in the membrane between new daughter cells upon midbody formation (**Fig 3E**)

We thank the reviewer for acknowledging the difficulties of imaging endogenous proteins This is particularly challenging at planes with higher depth, such as the midsagittal sections of *Drosophila* egg chambers used to image follicle cells along the apical-basal axis. We hope that by revising Figure 3, it has become clearer that *Dys:GFP* and *Dg-GFP* accumulate in the furrowing membrane, just below the contractile ring, and so basally as cytokinesis progresses from basal to apical. We edited the text to make this point clearer, as paraphrased here:

“became locally enriched at the basal part of the membrane at the onset of constriction and accumulated at the ingressing membrane below the ring during cytokinesis” Although there is close contact between these membranes and the basal part of the ring, we have no direct evidence that it is “over the contractile ring” (a similar conclusion can be taken from observing the notum (**Fig EV4C, EV4D**)). After ring closure, we detect the DAPC complex at the cell-cell contacts along the new interface formed between daughter cells, and also close to the midbody as observed in the kymographs of surface projections (**Fig. 3D, 3E**) and in the

separate channels of longitudinal projections (**Fig 3B, 3C**). To accommodate the separate channels and the new panel 3E, we have split Figure 3 into two figures.

3. *The redistribution of Dysshort:sfGFP and Dyslong:sfGFP to the contractile ring suggests it is independent of actin binding; is this redistribution dependent upon Anillin?*

Response: We thank the suggestion to search for possible alternative mechanism controlling Dys localization. Our experiments suggested that Dys:sfGFP localization is not affected by Anillin RNAi depletion (**Figure Rev3**). As there is no expected physical link between Anillin and Dystrophin, and since during constriction Dystrophin/Dystroglycan show enrichment in basal parts of the ingressing membrane that do not contain Anillin (e.g. Fig 3A), we did not feel it would be necessary to include this data in the manuscript. However, if the reviewer finds it useful, we will do so.

Figure Rev3: *Dys-GFP maintains its enrichment in the ingressing furrow below the cytokinetic ring in AniRNAi cells. Arrow marks the basal part of the ring labelled with Sqh:mKate in follicle cells depleted of Anillin by RNAi. Control: *tj-Gal4/Sqh:mKate2; Dys:sfGFP/+* (n=10). Anillin RNAi: *tj-Gal4, UAS-Anillin RNAi/SqhmKate2; Dys:sfGFP/+* (n=6). Scale bars: 5 μm*

4. *The cytokinesis failure in lines just expressing the short or the long isoform of Dystrophin is clear and impaired contractile ring ingression in the absence of dystrophin or dystroglycan is clear, but how are these proteins contributing to these processes - are there any elements within Dys that are needed for cytokinesis?*

Response: Since both isoforms of Dystrophin are required for cytokinesis, we hypothesized that the elements of Dys required for cytokinesis map to a region common to long and short isoforms. The Dg-binding domain plays a role in cytokinesis because Dg is required as a general membrane recruitment factor for Dystrophin (PMID: 16943280). There are two other well conserved domains shared by short and long Dystrophin that may play a role during cytokinesis: a conserved N-terminal spectrin repeat (SR24) that is proposed to interact with

microtubules (PMID: 19651889), and widely conserved motifs in the C-terminal part that interact with the adaptor proteins syntrophins and dystrobrevin (PMID: 11069112, PMID: 9356463). To test whether we could investigate the function of these domains while addressing their impact in Dystrophin localization, we generated transgenic flies expressing GFP-tagged versions of short Dystrophin isoform deleting them. Our data (**described in lines 267 to 280 of revised manuscript**) suggest that the cytokinetic redistribution of Dys is only disrupted by removing C-terminal region containing the syntrophins and dystrobrevin binding sites. In addition, this mutant version also has defects in cortical localization that could complicate the interpretation of other functional experiments at this stage. Although the characterization of a putative role of these proteins or related adaptor proteins that bind to the same motifs (a region containing an alpha-helix and adjacent coiled-coil domains) will be an interesting avenue for the molecular dissection of Dys function during cytokinesis, this would be impossible to accomplish in the time frame of a manuscript revision. We have included these new experiments as supplementary data (**Fig EV5C-G**), as they pave the way for future efforts beyond the scope of this research report. Moreover, we have mentioned in the discussion that these motifs could participate in the alternative connection between the DAPC and the cytoskeleton “Although Dys cytokinetic function require isoforms lacking known actin-binding domains, adaptor proteins that interact with conserved Dys C-terminal motifs could link the DAPC to the cytoskeleton”

5. *Is the GFP-tagging of Dystrophin functional? Given that neither *Dys^{short}:sfGFP* nor *Dys^{long}:sfGFP* can support cytokinesis in the absence of Anillin, is it possible that these are functionally compromised fusion proteins?*

Response: We should clarify that we have not tested if *Dys^{short}:sfGFP* or *Dys^{long}:sfGFP* can support cytokinesis completion in the absence of Anillin. Instead, we have used the untagged versions of the isoform-specific mutant alleles (*Dys^{long181}*, which mutates all the long isoforms including RH, and *Dys^{RE225}*, which mutates the short isoform RE).

We make reference to the difference between these alleles in the manuscript “we generated by CRISPR/Cas9 isoform-specific indel mutant alleles in an **untagged** *Dys* genomic locus (*Dys^{long181}*, i.e. mutating all long isoforms including RH, and *Dys^{RE225}* mutating RE) and in a **sfGFP-tagged** one. Naming of *Dys^{long181}:sfGFP* and *Dys^{RE225}:sfGFP* was simplified by indicating the tagged isoforms: *Dys^{short}:sfGFP* and *Dys^{long}:sfGFP*, respectively.” We understand the source of confusion as in the original version we have not directly stated in the text that the untagged alleles were used to evaluate cytokinesis defects. We therefore revised the text to mention this point “Accordingly, *Dys* disruption with either **isoformspecific untagged mutant alleles (*Dys^{long181}* and *Dys^{RE225}*)** dramatically increased multinucleation upon Anillin depletion (Fig 4C, D)”

Minor

L188. A call out to Fig 1F would help

Response: We have included this.

The renaming of the isoform-specific replacements as *Dys^{short}:sfGFP* and *Dys^{long}:sfGFP* is helpful, but this renaming seems only to have been applied to the main text - including these labels on Figures 3 and 4 markup would help readability.

Response: This point is related to the previous comment 5. As mentioned above, we use two types of isoform-specific alleles:

- The GFP-tagged mutant versions, which have been renamed to the available isoform that is tagged. These are used in the **Fig 4A, 4B** (previously Fig 3E-3F) to report localization.
- The mutant versions (untagged), which were named after the version that was mutated (long181 - stop at aa 181 of the long isoform, RE225 - stop at aa 225 of the short RE isoform) and used to evaluate cytokinesis defects, are shown in **Fig 4C, 4D, 5C, 5F, 5G** - (previously Fig 3G, 3H, 4C, 4F and 4G).

As mentioned in the response to point 5, we have revised the text to clarify the use of untagged alleles in the analysis of cytokinesis defects and welcome any further suggestions clarify this point.

Reviewer #2

Cytokinesis of the epithelial cells is not trivial since the cell-cell adhesion needs to be remodelled without affecting the barrier function of the tissue. In this manuscript, the authors performed an RNAi-based targeted screen for the modifiers of the cytokinesis failure due to the depletion of anillin, a widely conserved multifunctional linker protein crucial for cytokinesis. Cell-adhesion molecules such as cadherins and immunoglobulin-superfamily molecules, extracellular matrix proteins, and their receptors were examined. Among the positive hits, the authors focused on dystrophin-associated protein complex (DAPC). Deficiency of dystrophin (dys), light-induced artificial aggregation of Dys, and specific depletion of short or long isoform all enhanced the cytokinesis failure by anillin RNAi. Live observation revealed that dystrophin isoforms are enriched to the ingressing membrane during ring constriction and that dystrophin mutants (+anillin RNAi) show late cytokinesis phenotype. Delayed ring constriction was also observed in the dystrophin and dystroglycan mutants.

This work demonstrated that the DAPC contributes to the robustness of cytokinesis in epithelial tissues. It is suitable for publication in EMBO Reports after addressing a few points below.

Major points

1. Specificity of the role of DAPC in the epithelial cytokinesis.

Although the authors performed the screening and further analysis in the epithelial tissues, it is left unclear whether the role of DAPC for robust cytokinesis is specific to the epithelial tissues or not. Testing the enhancement of the anillin phenotype in non-epithelial tissues would strengthen the conclusion. Otherwise, it should be clearly stated that the role of DAPC might not be specific to epithelial cytokinesis.

Response: To explore the potential role of the DAPC beyond epithelia, we tested whether its cytokinetic redistribution would be observed in 2 non-epithelial systems that are classically used to study cell division in *Drosophila*: S2 cell culture and neuroblasts.

We could not detect Dys:sfGFP in dividing neuroblasts, which is consistent with its very low expression level as measured by RNAseq in neuroblasts (PMID: 22884370). We have also expressed both short and long isoforms of Dys in S2 cells, and none of them displayed any local redistribution during cell division (**Fig EV4A, EV4B**). Dg has moderate to high expression levels in S2-DGRC cells (PMID: 21177962), and so the inability of Dys to localize to the cortex is unlikely to reflect the absence of Dg binding, but rather the absence of ECM in S2 cell culture. So, we believe that the mechanism at work in cytokinesis is likely to be particularly relevant in contexts where DAPC is connected to the ECM, such as in epithelial cells.

As we could not detect any relevant cytokinetic enrichment in these non-epithelial contexts, we did not further study genetic interactions. However, we cannot rule out functions in other

non-epithelial cell types. In fact, Dys cytokinetic localization has been previously observed in rat embryonic fibroblasts and unpolarized cell culture (PMID: 18054267). We reinforce this point in the revised manuscript to make it clear that the role of the DAPC may not be specific of epithelial cytokinesis: “The cytokinetic redistribution of these proteins is not common to all cell types, as the main Dys isoforms do not show any cortical accumulation during cytokinesis in *Drosophila* S2 cells (Fig EV4B). Nevertheless, the DAPC may have cytokinetic functions beyond epithelia, as Dys and Dg showed cleavage furrow enrichment in rat fibroblasts and non-polarized cell culture models (Higginson et al, 2008; Villarreal-Silva et al, 2011). “

2. Variation of the control in the screen. Figs. 1D-G show that the spread of nuclei/cell in the mCherry RNAi + anillin RNAi strain varies from experiment to experiment. Its mean and median are also likely to vary. However, plotting the data by "Δ Multinucleation Ratio" prohibits the readers from checking this. Simply plotting 'nuclei/cell' would be more informative. At least, two columns for the mean and standard deviation of the mCherry control should be added in Table EV1.

Response: The reviewer is absolutely correct that the mean/median of the control can vary between experiments. When we initially designed the genetic screen, we noticed the variability in phenotype severity for Anillin RNAi alone. This motivated us to perform parallel control experiments in all experimental conditions and independent replicates and plot the variation of multinucleation as the relevant value to compare all different genetic manipulations.

We have followed the reviewer suggestion and now include in Table EV1 the statistics for the corresponding UAS-mCherry control used in all experimental conditions. We would like to note that readers will also have access to all source data used for quantifications, which will be included in the revised version of the manuscript.

3. Timing of furrowing

In Figs 4D~G, no difference was found in the rate of furrow constriction. The observed difference in the timing strongly depends on how the timing of the onset of furrowing (t=0) is defined. In the bottom row of Fig. 4E, there is no visible change of the diameter in the first 2 min after "constriction onset". An independent and more reliable standard such as anaphase onset by observing chromosomes is necessary to demonstrate the rather subtle difference.

Response: “*In Figs 4D~G, no difference was found in the rate of furrow constriction*”.

In Figure 4G (**revised to Fig 5G**), we have measured the ring constriction rate during the constant phase of ring constriction and observed a significant reduction in *Dg* and different *Dys* mutant alleles. This measurement is independent of how the time of furrow onset is defined, as it focuses solely on the linear phase of ring constriction and is not normalized to

the onset of constriction (it is shown in $\mu\text{m}/\text{min}$). We have improved the description of the methods to make this point clearer “To measure constriction rate during the constant phase of constriction ($\sim 80\%$ - 40% of the original diameter), we selected four consecutive time points for which the change in absolute ring diameter could fit to a linear regression ($R^2 > 0.95$ was set as the threshold for the linear fit to assume constant constriction rate). We determined the constriction rate (as the slope (β) of a linear regression ($d = \alpha + \beta t$ computed with Prism (GraphPad Software)).”

Nevertheless, we agree with the reviewer on the importance of defining the timing of the onset of furrowing ($t = 0$) for proper comparison of the data in Figure 4D, 4F (revised to **5D**, **5F**). We apologize for missing this definition in the previous version of the manuscript. It has now been included in the methods “normalized to the length of the ring at t_0 (defined as the frame prior to a change in ring diameter) and”. This initial change in ring diameter reflects the onset of furrowing and is independent from potential effects on anaphase duration, which could confound measurements that used anaphase onset as reference point. We note that during revision, we have reanalyzed the data to confirm that the reference time was properly determined for all measured cells.

Following the comment “*In the bottom row of Fig 4E, there is no visible change of the diameter in the first 2 min after "constriction onset"*”, we noticed a detail in Figure 4E (**Fig 5E** in the revised version) that could have caused the doubts on the definition of furrowing onset: the word **cytokinesis** on the top right at the start of the sequential projection could mislead the reader into expecting cells to constrict since the beginning of the projection. This word has now been removed as the first frames of the sequence are not yet in cytokinesis. We maintained the pre-constriction frames in the image to allow the reader to see a few frames prior to initiation of constriction (labelled by the bottom arrow “**constriction onset**”). Observing the whole ring presented in the manuscript in movie EV6 is an alternative way to appreciate the first contraction.

Finally, when revising the original data and quantification to ensure that the furrowing time was correctly determined, we noticed that we misplaced the beginning of constriction in the top and bottom part of panel E. We therefore adjusted them in the revised version and the related movie EV6.

Minor points

(Fig. 1D~G legend)

Δ Multinucleation Ratio = (nuclei/cell)RNAi - mean (nuclei/cell)mCherry
(line 533 Materials and Methods)

(nuclei/cell)cell adhesion RNAi - $(1/n) * \Sigma(x \text{ nuclei/cell})$, control, where n represents the number of control egg chambers within each replicate. What is x)?

Response: We thank the reviewer for pointing this out. The x was there by mistake and has been removed.

Fig. 2F and G missing control for the non-specific effect of light exposure. For example, 72 h light exposure of the Dys:sfGFP VHH:CY2 stain without CIBN:MP should not show cytokinesis failure.

Response: We appreciated the suggestion of this important control. In the LARIAT setup that we used, CRY2:VHH and CIBN:MP is cloned with the 2A linker system to drive simultaneous expression of both constructs, and so we cannot express one without the other. We therefore simply co-expressed UAS-Anillin RNAi and the UAS-LARIAT system to test if light exposure could affect the multinucleation rate in egg chambers that lacked Dys:sfGFP clustering. Surprisingly, there was a very significant increase of multinucleation with our light exposure method (**Figure Rev 3, condition 2 vs 4**). This could suggest that blue light interfered with cytokinesis completion, but this is unlikely since blue-light induced clustering alone did not cause multinucleation (**Figure Rev3 – experimental condition 1**). In addition, we found that even though the optogenetic box was placed in a 25°C incubator, the heat generated when the light was turned on led to a local increase in temperature (temperature inside the fly vial stabilized at ~29°C during light exposure as measured by a digital air thermometer). It is established that expression driven by the UAS/GAL4 system is enhanced at 29 °C in comparison to lower temperature (PMID: 7707973). So, it is anticipated that there is increased expression of UAS-driven Anillin RNAi in relation to the dark controls placed inside the 25 °C incubator, but outside the optogenetic box. Thus, one likely explanation for the increase in multinucleation rate would be an increased efficiency of RNAi-mediated depletion of Anillin caused by higher temperature. Consistent with this, flies incubated in vials inside the blue-light optogenetic box to raise the temperature, but covered by cardboard to prevent light exposure also showed a higher frequency of multinucleated cells (**Figure R3, experimental conditions 2 vs 3**). Altogether, we concluded that temperature increase inside the optogenetic box rather than blue-light exposure has been a confounding factor in these experiments.

We thank the reviewer for guiding us into this essential experimental control. We should note that the multinucleation frequency is higher in homozygous Dys:sfGFP follicle cells with light-induced clustering than in follicle cells simply expressing Anillin RNAi under blue light (**Figure Rev3, experimental condition 4 vs 6**). However, the identification of a confounding factor caused by long-term incubation in the optogenetic box led us to remove the set of experiments that involved light-induced clustering from the manuscript.

The LARIAT experiment was originally conducted to further test the phenotypic interaction between Dys and Anillin and to confirm that the increased frequency of multinucleated cells resulted from cytokinesis defects during proliferative stages. We consider that removing these data will not weaken the overall conclusions of this manuscript. First, the interaction between Dystrophin and Anillin is supported by complementary experiments using four different mutant alleles (Fig 2B, 2C and Fig 4C, 4D). Second, we directly demonstrated the role of the DAPC in cytokinesis through live imaging of cell division. This imaging analysis revealed: 1) a consistent relocalization of the DAPC complex associated with cytokinesis during the proliferative stages of the follicular epithelial and other epithelial tissue (Fig 3, Fig 4B, Fig EV4C, EV4D); 2) the impact of DAPC mutants on cytokinesis failure and ring constriction during cytokinesis (Fig 5). These results robustly support the involvement of DAPC in cytokinesis, making the main conclusions well-supported even without the LARIAT data.

Figure R3: Analysis of non-specific effects produced by long term incubation in the optogenetic box with light.

Plot shows percentage of multinucleated cells in stage 10 egg chambers dissected upon 72h incubation in the indicated experimental conditions.

Exp. condition 1: (light, no RNAi: *tj-Gal4/UAS-LARIAT*);

Exp. Conditions 2-4 (*tj-Gal4*, UAS-Anillin RNAi/UAS-LARIAT);

Exp. conditions 5-6 (*tj-Gal4*, UAS-Anillin RNAi/UAS-LARIAT; Dys:sfGFP).

Mean is indicated: *p*-value calculated by nonparametric unpaired Mann-Whitney test.

Cytokinesis, the process of separating daughter cells after cell division, can be challenging for epithelial cells. These cells are connected to neighboring cells and the extracellular matrix through transmembrane protein complexes. To understand how cell adhesion machinery affects cytokinesis efficiency, Gonçalves and cols conducted an RNAi-based study in the *Drosophila* follicular epithelium. They found that certain adhesion molecules and transmembrane receptors aid cytokinesis completion. They show that Dystrophin/Dystroglycan, which link the extracellular matrix to the cell's cytoskeleton, concentrate in the ingressing membrane below the cytokinetic ring during and after constriction. Functional analyses let the authors suggest that the machinery involved in cell-cell and cell-matrix interactions doesn't hinder cytokinesis; instead, it has evolved to ensure efficient cell division in epithelial tissues.

The findings presented here shed light on how cell-cell and cell-matrix interactions contribute to efficient cell division in epithelial tissues. Overall, this work provides valuable insights into cellular processes and has broader implications for understanding tissue development and, potentially, health. This study is well presented, written, and organized, and holds general interest.

I would however suggest addressing some specific points (see below), before the MS is ready for publication:

The authors use a cytokinesis-sensitized background produced by RNAi against Anillin. It would be interesting to test if the effects observed by Dystrophin/Dystroglycan (over-expression and depletion) is specific for that background or can also modify the effects of other elements involved in cytokinesis other than anillin. These experiments would show if the effect is specific for anillin depletion or has a more general role in the cytokinesis process.

Response: We thank the reviewer for this comment (shared with reviewer 1), which prompted us to test whether Dystrophin/Dg function would affect other backgrounds sensitized to cytokinesis failure by tissue-specific depletion of 3 genes involved in different aspects of cytokinesis:

- **Tumbleweed** (RACGAP1 in mammals) is a component of the centralspindlin complex that acts during cytokinesis from ring assembly to midbody formation and abscission (PMID: 22927365). We found that Tum RNAi alone causes dramatic cytokinesis defects in the follicular epithelium. We thereby used the GAL80ts module under 2 days of incubation at 29°C (3 days were used in the screen) to further restrict the period of RNAi expression. Under these conditions, Tum RNAi still produced strong cytokinesis defects (11 % of egg chambers (n = 62) showed all follicle cells multinucleated; mean frequency of multinucleated cells = 68,3 %). Importantly, there is a significant enhancement of multinucleation by co-expression of Dg RNAi (**Fig EV3D**, 23 % of egg chambers (n = 73) showed all follicle cells multinucleated; mean frequency of multinucleated cells = 82,4 %, p = 0,0027).

- **Rok** (ROCK in mammals) is involved in contractile ring formation and constriction (PMID: 16488869). In contrast to Tum RNAi, follicle cell-specific depletion of Rok induced only

very weak multinucleation defects (48% of egg chambers (n = 50) **without** any multinucleated cells; mean frequency of multinucleated cells = 0.82%). This weak phenotype is not ideal for evaluating genetic interactions, as the efficiency of cytokinesis completion may not be properly challenged under these conditions. Nevertheless, we observed a slight increase in multinucleation frequency by co-expressing Dg RNAi (**Fig Rev1**, 19 % of egg chambers (n = 52) **without** any multinucleated cell; mean frequency of multinucleated cells = 1, 34 %, p=0.023). Given the very weak multinucleation phenotype under these conditions, we did not consider this result compelling enough to include in the manuscript.

- The Septin **Peanut** (Septin7 in mammals) is required for cytokinesis by forming Septin filaments that associate with Anillin and help to attach the contractile ring and subsequently the midbody ring to the membrane (PMID: 8181057, PMID: 22804577). Unfortunately, we were unable to test a possible interaction between this Septin (RNAi line: P(TRiP.HMC05924)attP40 - BL accession # 65157) and the DAPC complex (Dg RNAi or Dys mutants), because Pnut depletion caused severe defects in the morphogenesis of egg chambers, which failed to reach stage 10 of oogenesis. For this experiment, we generated flies with the following genotypes: Pnut RNAi (tj-Gal4/UAS-Pnut RNAi; UAS-mCherry RNAi/+) vs. Pnut RNAi +Dg RNAi (tj-Gal4/UAS-Pnut RNAi; UAS-Dg RNAi/+) or Pnut RNAi (tj-Gal4/UASPnut RNAi) vs Pnut RNAi + Dys mutants (tj-Gal4/Pnut RNAi; Dys MI025024/DysDf), and all of them lacked stage 10 egg chambers.

We were unable to perform similar interaction analysis between Tum RNAi and Dg overexpression. The UAS-Dg transgene is placed in the second chromosome, which prevented the use of the same genetic strategy employed to test the interaction with UAS-DgRNAi (located on the third chromosome).

In conclusion, we included in the manuscript the interaction analysis between Tum RNAi and Dg RNAi (**Fig. EV3D**), to provide evidence that DAPC can promote cytokinesis efficiency under different conditions that sensitize cells for cytokinesis failure, further supporting a role in cytokinesis that goes beyond Anillin function. Related to this, we also included data supporting that Anillin recruitment to the cytokinetic ring is not affected in Dys mutants (**Fig. 2D, 2E**).

In Fig 1B, C, the authors should show if UAS-cad, on its own, has an effect in cytokinesis, independently of Anilin-RNAi. This control is needed to better interpret their results.

Response: We have now tested whether overexpression of ECad alone is sufficient to induce cytokinesis failure. The data in **Figure 1C** shows that this is not the case. Thus, this result highlights that increased adhesion interferes with cytokinesis efficiency, but is insufficient on its own to induce cytokinesis failure.

The authors provide some analysis of RNAi efficiency by immunofluorescence, but this is only limited to some of the genes analyzed. It would be important to use those techniques or qPCR in the other conditions analyzed. This would validate the use of those tools and could allow to present that lack or presence of phenotypes with more confidence.

Response: In the original version of the manuscript, we focused on validating RNAi protein depletion when we were taking conclusions from absence of phenotypic modification, namely ECad, Integrin, Collagen IV and Laminin (Fig EV2A-D in previous version). Although we originally prioritised this set of validations, we agree that it would be important to be able to assess depletion efficiency for more genes from the screen. We have not attempted to use qPCR, as the expression of some of the genes we depleted in the somatic follicular epithelium is significant in other ovary cell types, such as the germline (where there is no depletion by the *tj>GAL4* driver). This could complicate the interpretation of qPCR results from whole ovaries. Instead, we have now validated the efficiency of protein depletion for RNAis against the major regulators of cell-cell and cell-matrix interactions (*new data for NCad, Echinoid, Neuroglian, Fas2, Fas3, Dg, Perlecan, LanB1*) by imaging egg chambers using antibodies or transgenic lines expressing endogenously tagged proteins. Together with previous data, the new **Fig EV2** shows the validation of RNAi depletion for the key components of cell-cell (ECad, NCad, Echinoid, Neuroglian, Fas2, Fas3) and cell-matrix interactions (Dg, Integrin, Perlecan, Collagen IV, Laminins (LanA, LanB1), and is referred to in **line 189**.

Furthermore, we should note that in order to confidently present the most relevant interactions with Anillin RNAi, we used alternative genetic perturbations (multiple RNAis for Perlecan and Neuroglian or several overexpression/mutants for DAPC), as these are useful to rule out potential off-target effects that could be present in a single RNAi line. Moreover, we have further validated some RNAis based on egg chamber elongation phenotypes described in the literature (Integrins, Kug and Collagen IV, **Fig EV3B**). Thus, and even if we did not manage to validate RNAi depletion for all lines, we have validated the RNAi lines used to draw the main conclusions in the paper.

In some panels, the authors use the "UAS" term to label the genotypes (eg, Fig 1C) in others they don't (Fig 1D). All those transgenes are UAS-driven. I would suggest being consistent with the nomenclature. Otherwise, it could be confusing to the reader. For example, in Fig 1F, the authors label a data set as UAS Dg-C, but the other transgenes (eg mCherry, or Dg RNAi) do not have the UAS in them. For consistency, I would suggest reviewing that."

Response: Indeed, all are UAS constructs, and we followed the reviewer suggestion to ensure consistency between RNAi and overexpression conditions.

We simply removed the word "UAS" from Fig 1B, Fig. 1C, Fig. 1F, and included OE (overexpression) to make clear the difference between all other UAS-driven RNAi and the UAS-driven overexpression.

Dear Eurico,

Thank you for submitting your revised manuscript. It has now been seen by all of the original referees. Please accept my apologies for this unusual delay in getting back to you, which was due to a combination of the delay in obtaining the referee reports and my conference travel.

As you can see, the referees find that the study is significantly improved during revision and recommends publication. However, I need you to address the points below before I can accept the manuscript.

- We note that two preprints were cited in the manuscript. Please revise their in-text citations and their citations in the reference similar to the example below:

In-text citation: (preprint: NAME1 et al, YEAR)

Author NAME1, Author NAME2, (YEAR) article title. bioRxiv doi: 1234/002.d.fj123 [PREPRINT]

- We note that the following funding information is missing from the manuscript tracking system:

- "FCT Scientific Employment Stimulus - Individual Call" program (CEECIND/00622/2017)

- PhD fellowship from FCT (SFRH/BD/130708/2017)

- IPATIMUP - CANCER_CHALLENGE2022

- i3S Scientific Platform ALM, member of the national infrastructure PPBI - Portuguese Platform of Bioimaging (PPBI-POCI-01-0145-FEDER-022122)

- Institut Curie PICT-IBISA@BDD imaging facility (member of the French National Research Infrastructure France-Bioimaging, ANR-10-INBS-04)

- the French government IDEX-ISITE initiative 16-IDEX-0001 (CAP 20-25)

- the Institut Curie

- CNRS

- INSERM

- ARC (SL220130607097)

- CANCERO-INCA (PLBIO2020/BELLAICHE)

- We note that Table EV1 was uploaded as a zip folder, but this is a dataset and it should be uploaded as a single Excel file named Dataset EV1; its legend should be provided in the Dataset Excel file as a separate tab/sheet; Table EV2 and Table EV3 should be removed from the manuscript and uploaded as separate files: Table EV1 and Table EV2

- As per the movies, their legends need to be removed from the manuscript file - each should be provided in a readme.txt file and then should be zipped up with its corresponding movie file so that we have one zip folder per movie uploaded.

- Our production/data editors have asked you to clarify several points in the figure legends:

- o Please note that the exact p values are not provided in the legends of figures 1c-e, g; 2b; 4c; 5d, f; EV 3b.

- o Please note that the error bars are not defined in the legend of figure EV 3b.

- o Please note that the scale bar needs to be defined for figure 2d.

- o Please note that the yellow box is not defined in the legend of figure 5e. This needs to be rectified.

- o Please note that the yellow arrowheads are not defined in the legend of figure 2d. This needs to be rectified. S

- Papers published in EMBO Reports include a 'synopsis' and 'bullet points' to further enhance discoverability. Both are displayed on the html version of the paper and are freely accessible to all readers. The synopsis includes a short standfirst summarizing the study in 1 or 2 sentences (max 35 words) that summarize the paper and are provided by the authors and streamlined by the handling editor. I would therefore ask you to include your synopsis blurb and 3-5 bullet points listing the key experimental findings.

- In addition, please provide an image for the synopsis. This image should provide a rapid overview of the question addressed in the study but still needs to be kept fairly modest since the image size cannot exceed 550 (width) x 300-600 (height) pixels.

Thank you again for giving us to consider your manuscript for EMBO Reports, I look forward to your minor revision.

Kind regards,

Deniz

--

Deniz Senyilmaz Tiebe, PhD

Senior Scientific Editor

EMBO Reports

Referee #1:

I appreciated the authors efforts to address my points and their explanations. I'm happy to recommend publication. Thanks

Referee #2:

In this revised version, the authors have properly addressed all the points I raised regarding the original manuscript. Notably, they performed a control experiment for the LARIAT technology and chose to withdraw the results obtained through this method. I agree with their decision and find that the main conclusions are still well-supported by other data.

Referee #3:

The authors have addressed my concerns and the paper is suitable for publication.

All editorial and formatting issues were resolved by the authors.

Dr. Eurico Morais-de-Sá
i3S - Instituto de Investigação e Inovação em Saúde
Epithelial Polarity and Cell Division
Instituto de Biologia Molecular e Celular
Universidade do Porto
Porto, Rua Alfredo Allen, 208 4200 - 135 Porto
Portugal

Dear Eurico,

Thank you for submitting your revised manuscript. I have now looked at everything and all is fine. Therefore, I am very pleased to accept your manuscript for publication in EMBO Reports.

Congratulations on a nice work!

I need your input on one more point before we can export your manuscript to our production team. I am afraid the text in the synopsis image is not easy to read currently. Therefore, I would like to ask you to send me a version where the font colors of the synopsis image are darker. You can send me the file per email. Thank you.

Kind regards,

Deniz

--

Deniz Senyilmaz Tiebe, PhD
Senior Scientific Editor
EMBO Reports
